# Baicalein Inhibits Stx1 and 2 of EHE: Effects of Baicalein on the Cytotoxicity, Production, and Secretion of Shiga Toxins of Enterohaemorrhagic *Escherichia coli*

**DOI:** 10.3390/toxins11090505

**Published:** 2019-08-29

**Authors:** Pham Thi Vinh, Yui Shinohara, Akifumi Yamada, Hoang Minh Duc, Motokazu Nakayama, Tadahiro Ozawa, Jun Sato, Yoshimitsu Masuda, Ken-Ichi Honjoh, Takahisa Miyamoto

**Affiliations:** 1Division of Food Science & Biotechnology, Department of Bioscience and Biotechnology, Faculty of Agriculture, Graduate School, Kyushu University, 744 Motooka, Nishi-ku, Fukuoka 819-0395, Japan; 2Global R&D-Safty Science, Kao Corporation, 2606, Akabane, Ichikai-machi, Haga-gun, Tochigi 321-3497, Japan; 3Bioscience Research, Kao Corporation, 2606, Akabane, Ichikai-machi, Haga-gun, Tochigi 321-3497, Japan

**Keywords:** baicalein, Stx, inhibition, cytotoxicity, enterohaemorrhagic *Escherichia coli*

## Abstract

Shiga toxin-producing enterohaemorrhagic *Escherichia coli* (EHEC) O157:H7 is an important foodborne pathogen. Baicalein (5,6,7-trihydroxylflavone), a flavone isolated from the roots of *Scutellaria baicalensis*, is considered as a potential antibacterial agent to control foodborne pathogens. Among seven compounds selected by in silico screening of the natural compound database, baicalein inhibited the cytotoxicity of both Shiga toxins 1 and 2 (Stx1 and Stx2) against Vero cells after pretreatment at 0.13 mmol/L. In addition, baicalein reduced the susceptibility of Vero cells to both Stx1 and Stx2. Real-time qPCR showed that baicalein increased transcription of *stx1* but not of *stx2*. However, baicalein had no effects on production or secretion of Stx1 or Stx2. Docking models suggested that baicalein formed a stable structure with StxB pentamer with low intramolecular energy. The results demonstrate that inhibitory activity of baicalein against the cytotoxicity of both Stx1 and Stx2 might be due to of the formation of a binding structure inside the pocket of the Stx1B and Stx2B pentamers.

## 1. Introduction

Enterohaemorrhagic *Escherichia coli* (EHEC) causes foodborne illness and can lead to haemorrhagic colitis (bloody diarrhoea) and potentially fatal haemolytic uraemic syndrome (HUS) [1]. Shiga toxin (Stx) is an important virulence factor of EHEC, also known as verocytotoxin or vero toxin. There are two subgroups of Stx, namely Stx1 and Stx2, which are found in different combinations in EHEC isolates [2]. Stx is one of the AB-5 family of toxins, consisting of a pentameric B subunit noncovalently bound to an enzymatically active A subunit. The Stx receptors are glycolipids of the globo-series, of which globotriaosylceramides (Gb3s) is the primary receptor found on the surface of vascular endothelial cells and kidney epithelial cells. However, Stx subtypes have been shown to bind with different affinities to multiple glycolipid receptors. Sx1a preferentially binds to Gb3, with detectable binding to globotetraosylceramide (Gb4) whereas Stx2a strongly binds to Gb3 and marginally binds to Gb4 [3]. Stx2e binds to both Gb3 and Gb4, and also to the Forssman glycosphingolipid [4]. The Stx pentameric B subunit interacts with Gb3 and induces membrane invagination, leading to the internalization of the toxin. The internalized Stxs inhibit protein synthesis, leading to cell death. Recently, however, Lee et al. (2019) reviewed the multifunctionality of Stxs, which only inhibits protein synthesis but also induces apoptosis in different cell types [5].

Bacterial infections are commonly treated with antibiotics. However, antibiotic therapy in the treament of EHEC infection is still controversial because the increase in Stx production and secretion leads to the risk of HUS development [6,7,8]. Certain drugs commonly used in the clinic to treat EHEC-infected patients are not recommended, such as antibiotics, antimotility agents, narcotics, and non-steroidal anti-inflammatory agents [9]. In addition, Hiroi et al. (2012) indicated that EHEC strains isolated from humans have shown increased resistance to one or variety antibiotics [10]. Clearly, the demand for the development of novel therapies to prevent or treat EHEC infections is increasing.

Plant polyphenols and plant extracts, including polyphenols, have been reported to inhibit cholera [11] and Stx2 toxins [12], among others. Furthermore, polyphenols have almost no toxicity to humans and some natural phenolic compounds could be developed as effective drugs. For example, the extracts of plants and mushrooms are cost-effective alternatives for antibiotics to enhance innate immunity in chickens [13]. Phytochemicals, including plant polyphenols, have shown inhibitory effects on verocytotoxin-producing *E. coli* O157:H7 [14]. The combination of phenolic compounds and antibitotics has also shown synergistic effects against both *Staphylococcus aureus* and *Escherichia coli* [15]. Further studies in this area would be helpful to develop the new methods for the treatment of bacterial infections. 

We previously investigated the inhibition effects of green tea polyphenols on Stxs at low concentrations using purified catechins. In the previous study [16], we showed that the cytotoxicity of Stx1 was reduced by pretreatment with gallocatechin gallate (GCg) and epigallocatechin gallate (EGCg). In contrast, catechins and theaflavin did not inhibit the cytotoxicity of Stx2 [16]. These results suggest that the tertiary structure of gallocatechin, which is from the galloyl group, is important for Stx1 cytotoxicity inhibition. 

In this study, we examined the effects of phenolic compounds, screened by docking simulation from the natural compounds database, on the cytotoxicity of Stx. Among the compounds selected by the in silico screening, baicalein reduced the cytotoxicity of Stx1 and Stx2. The effects of baicalein, which showed the strongest inhibitory activity against Stx, were investigated on the protection of Vero cells against Stx, and the production of Stxs by EHEC.

## 2. Results

### 2.1. Effects of Baicalein on the Cytotoxicity of Stx

The natural compounds selected from the in silico screening are listed in Table 1. The Stx was preincubated with or without each of the compounds at 100 mg/L before addition to the Vero cells culture. After incubation at 37 °C for 24 h, the viability of Vero cells was determined. The cytotoxicity of Stx1 and Stx2 decreased after preincubation with baicalein, in contrast to preincubation with the other compounds (Appendix A). 

The effects of baicalein on the cytotoxicity of Stx1 and Stx2 were further investigated in detail. Baicalein was mixed with the Stx preparations with different Stx concentrations and incubated for 1 h. The mixture was then added to the Vero cell culture. Figure 1 shows the effects of baicalein on the cytotoxicity of Stx1 and Stx2. The viability of Vero cells decreased with increasing concentration of Stx1 and Stx2, both in the absence and presence of baicalein. However, the cytotoxicity of both Stx1 and Stx2 was significantly reduced (*p* < 0.01) by the pretreatment with baicalein (Figure 1A,B). At 0.13 mmol/L, baicalein significantly reduced the cytotoxicity of Stx1 at concentrations ranging from 0.5 to 33.3 ng/mL, and to that of Stx2 from 2.1 to 533.3 ng/mL. The results clearly demonstrate that baicalein inhibited the cytotoxicity of both Stx1 and Stx2. 

### 2.2. Protective Effect of Baicalein on Vero Cells against Stx

To determine the protective effects of baicalein on Vero cells against Stx, Vero cells were preincubated with baicalein and then Stx preparations were added to the cells. Figure 2 shows the effects of Stx preparations on the viability of Vero cells pretreated with baicalein. In the absence or presence of baicalein, increasing the concentration of Stx1 and Stx2 decreased the viability of Vero cells. However, even in the presence of Stx1 or Stx2, the viability of Vero cells pretreated with baicalein was significantly higher than that of control without the pretreatment (Figure 2A,B). It seems that baicalein protected Vero cells from the cytotoxicity of both Stx1 and Stx2.

### 2.3. Effects of Baicalein on Production of Stx by EHEC

The transcriptional levels of *stx* were evaluated by real-time qPCR. The relative quantity of transcripts of *stx1* and *stx2* was compared between the cells treated with and without baicalein and MMC. Figure 3 shows the effects of baicalein and MMC on the transcription of *stx* genes. After treatment with baicalein, the transcription level of *stx1* was enhanced to 2.9-fold that of negative control (Figure 3A), whereas no significant difference in the transcription level of *stx2* was obtained (Figure 3B). However, in both the extracellular and intracellular samples prepared from *E. coli* O157:H7 treated with baicalein, amounts of both Stx1 and Stx2 were similar to those of the samples without baicalein treatment (Table 2). For the positive control, MMC induced the transcription level of both *stx1* and *stx2* by 17.8- and 6.8-fold compared to the negative control, respectively (Figure 3A,B).

To evaluate the effects of baicalein on the secretion of Stx in detail, cytotoxicity of the extracellular and intracellular Stx preparations was investigated. As shown in Figure 4, increasing the concentration of extracellular and intracellular Stx1 and Stx2 decreased the viability of Vero cells (Figure 4A,B). There was no significant difference in viability between Vero cells in the presence of Stx prepared from *E. coli* O157:H7 cultured in the presence and absence of baicalein. The results suggest that baicalein had no effects on the secretion of both Stx1 and Stx2.

### 2.4. Interaction of Baicalein with Stx1B and Stx2B Pentamers

Figure 5 shows the docking models of baicalein bound to the pockets of Stx1B and Stx2B pentamers. According to the conformation of the Stx1B and Stx2B pentamers, the potential site for binding to baicalein was estimated to be from Trp33 to Gly46 (Trp-Asn-Leu-Gln-Ser-Leu-Leu-Leu-Ser-Ala-Gln-Ile-Thr-Gly) in the Stx1B monomer and from Trp32 to Gly45 (Trp-Asn-Leu-Gln-Pro-Leu-Leu-Leu-Ser-Ala-Gln-Leu-Thr-Gly) in the Stx2B monomer [17,18]. A-E and A-J were respectively named for each the monomers of Stx1B and Stx2B pentamers [16]. The results show that baicalein bound to the side chains of amino acids facing inside the pocket of the Stx1B pentamer by forming two hydrogen bonds at ①Ser42 and ②Ser42 of Monomer B (Figure 5A). Similarly, for the case of Stx2B pentamer, baicalein formed 1 hydrogen bond at ①Ser41 of Monomer J (Figure 5B). The lowest intramolecular energy of the bonds of baicalein with the Stx1B and Stx2B pentamers was 1.8 and 0.2 kcal/mol, respectively. These results indicate that baicalein formed a complex structure with both the Stx1B (pocket size: 778Å^3^) and Stx2B pentamers (pocket size: 475Å^3^).

## 3. Discussion

Baicalein is a major component of *Scutellaria baicalensis*, a Chinese herb. In addition to its inhibitory effect on lipoxygenase and reverse transcriptase, it was found to have antioxidant, neuroprotective, antibacterial, antiviral, and antifungal activities [19,20,21]. Baicalein has also been reported to inhibit the biofilm formation of *Candida albicans* [22], and to induce apoptosis in a variety of human cancer cell lines [23,24,25,26]. Baicalein has been reported to induce cancer cell death and proliferation retardation by inhibiting CDC2 kinase and survivin [27]. Baicalein has been shown to inhibit enzymes of the cytochorome P450 system (CYP2C9) which metabolizes drugs in the body by binding to the substrate site of the enzyme [28]. Furthermore, baicalin, a parental compound of baicalein, significantly reduced the activity of Stx2 which induced lethality in mice [29]. The protection effects of baicalin on the cell against EHEC infection were clarified by Zhang et al. (2017) [30]. To the best of our knowledge, however, there are currently no reports on the inhibition effect of baicalein on the cytotoxicity of both Stx1 and Stx2. Therefore, further research with purified Stxs is needed to confirm the results of the in silico study. This is the first report that demonstrates that baicalein inhibited the cytotoxicity of both Stx1 and Stx2, possibly by forming a binding structure inside the pocket of the Stx1B and Stx2B pentamers.

The extent of the inhibitory activity of baicalein against the cytotoxicity of Stx1 was similar to that of EGCg, which did not inhibit the cytotoxicity of Stx2 [16]. Docking simulation suggests that baicalein formed two hydrogen bonds with amino acids inside the pocket of the Stx1B pentamer, with the low intramolecular energy of 1.8 kcal/mol, but it formed only one hydrogen bond with the low intramolecular energy of 0.2 kcal/mol with the Stx2B pentamer (Figure 5). It has been previously reported that 6 hydrogen bonds were formed in the interaction between EGCg and Stx2B pentamer, with the lowest intramolecular energy of 5.2 kcal/mol [16]. Since the molecular weight of baicalein (MW: 270.2) was much lower than that of EGCg (MW: 458.4), baicalein seemed to form a stable hydrogen bond with the side chain of amino acid facing inside the pocket of the Stx2B pentamer. This bond was formed with the lowest intramolecular energy when compared to the bonds of EGCg with the Stx2B pentamer even though the number of hydrogen bonds with the pentamer was lower than that of EGCg.

Furthermore, baicalein protected Vero cells from cytotoxicity of both Stx1 and Stx2. It has been reported that baicalein protected human hepatoma cells against attacks by cancer cells due to its ability to prevent the adhesion, migration, and invasion of cancer cells [31]. In addition, baicalein has been shown to protect human skin cells againts oxidative stress by Ultraviolet B (UVB) via preventing reactive oxygen species and absorbing UVB radiation [32]. Cariddi et al. (2015) reported that caffeic acid, a polyphenol, protected Vero cells from the cytotoxicity of ochratoxin A by altering principally the lysosomal function in Vero cells [33]. It seems that baicalein protects Vero cells from cytotoxicity of Stx by binding to the surface of the cytoplasmic membrane of the cell and altering the function of the membrane. At 0.13 mmol/L, the preincubation with bacialein reduced Vero cells sensitivity to Stx1 and Stx2 by approximately 6- and 8-fold, respectively (Figure 2). Cell protection against Stx of other compounds has also been reported in many studies. For exemple, the protective effect of 4 phenylacetyl-Arg-Val-Arg-4-amidinobenzylamide at concentrations of 25 µmol/L on Hep-2 cells againsts Stx increased by 6-fold [34], while chloroquine at concentration 25 µmol/L reduced Hep-2 cells sensitivity to Stx2 by 20-fold [35]. Although these compounds protected cells at the lower concentration, they have not been used in humans beacause of their toxicity [36,37].

The inhibition effects of baicalein on the cytotoxicity of Stx were determined (Figure 1 and Figure 2). As the results in Figure 1 and Figure 2 show, there was a decrease in inhibition of bacalein at 0.027 mmol/L in Figure 2 compared to the results in Figure 1. One of the possible explanations for this could be that the binding of Stx to the surface of Vero cell was more preferential than the binding of baicalein to StxB pentamers

In addition, baicalein significantly inhibited the expression of CTX-M-1 mRNA expression in *Klebsiella pneumoniae* strains [38]. We therefore performed experiments designed to determine the effects of baicalein on *stx1* and *stx2* transcription compared to negative (water) and positive (MMC) controls. In our study, treatment with MMC strongly increased the transcription of both *stx1* and *stx2*, while baicalein slightly increased the transcription of *stx1* but not *stx2* (Figure 3). Stx production of EHEC has been reported to be controlled by several factors, such as growth phase, reactive oxygen species, quorum sensing, H_2_O_2_, and neutrophils [39,40,41]. Specifically, Stx production by EHEC increase at the stationary growth [39,42]. It is also known that Stx production was regulated by phage through the amplification of gene copy number and toxin release [42]. Wagner et al. (2002) suggested that damage of bacterial cells could lead to the release of Stx by the absence of phage-mediated lysis in *E. coli* O26:H19 [43]. Moreover, baicalein inhibited growth of *Staphylococcus aureus* by the aggregation of bacterial cells and damaging the bacterial cell membrane [44]. In addition, it has also been reported that the increase in membrane fluidity was associated with the increase of Stx secretion in *E. coli* O157:H7 [45]. Wu et al. (2013) reported that baicalein exhibited antibacterial activity against *E. coli* by lowering membrane fluidity of the cells [46]. Together with the above findings, the results in this study suggest that baicalein did not affect the growth and membrane fluidity of *E. coli* O157:H7 at 0.38 mmol/L, though it reduced cytotoxicity of Stx and protected Vero cells at the lower concentration (0.13 mmol/L). However, our data showed that baicalein had no significant effects on the secretion of both Stx1 and Stx2 (Figure 4). This is an advantage of using baicalein to treat EHEC infections over antibiotics.

Some studies suggested that baicalein may be developed as a potential candidate for treatment of various disease models such as diabetes, cardiovascular diseases, inflammatory bowel diseases, gout and rheumatoid arthritis, asthma, encephalomyelitis, and carcinogenesis, neurological diseases, Alzheimer’s and Parkinson’s disease [47,48]. In 2019, Javed and Ojha reported that oral administration of baicalein for treatment Parkinson’s disease was safe for human [49]. It has also been reported that a mixture of baicalein in the dietary metabolic management of osteoarthritis was safe for human consumption [50]. Together with these facts, our results suggest that baicalein is one of the potential candidates for antivirulence strategies against EHEC infection. However, further research is still required to carefully evaluate the application of baicalein in the clinic. 

## 4. Materials and methods

### 4.1. Analysis of Interaction between StxB Pentamers and Natural Products

The interaction between the compounds in the MEGxp, the library of plant-derived natural products, and the Stx2 B subunit pentamer was analysed using the CDOCKER module of the Discovery Studio software (c41b1, Accelrys, Inc., San Diego, CA 92121, USA), on a Windows XP PC. The crystal structure of the Stx2 B subunit pentamer [17] registered in the Protein Data Bank as PDB ID: 3MXG was used in the calculations. Docking simulations were conducted with the compound positioned in the center of the Stx2 B subunit pentamer pore. Docking was performed by using molecular dynamics simulations. The conditions used for the screening of the natural products were as follows: the molecular weight was less than that of the EGCg (MW: 458.4), and the lowest intramolecular energy (strain energy + electrostatic energy) was less than 2 kcal/mol. The molecular structure of the binding between the B subunit pentamers and the compounds was determined using the free energy minimization method [51]. The compounds selected from the in silico screening were obtained from NAMIKI SHOJI Co., Ltd., Tokyo, Japan.

### 4.2. Preparation of Stx1 and Stx2

The preparations containing Stx1, Stx2 and no Stx preparations were respectively prepared from the cultures of *E. coli* O157:H7 No. 33 (*stx1*+, *stx2*−), O157:H7 No. 184 (*stx1*−, *stx2*+) and O157:H7 No. 37 (*stx1*−, *stx2*−), following the previously described procedure by Miyamoto et al. (2014) [16]. These *E. coli* O157:H7 strains were cultured in Luria Broth (LB, Becton Dickinson Company, Franklin Lakes, NJ, USA) at 37 °C for 24 h with shaking, then polymyxin B was added to the culture and incubated at 37 °C for 1 h. The final concentration of polymyxin B in the culture was 5000 U/mL. After centrifugation at 3300× *g* for 15 min at 4 °C of 25 mL of the culture, the supernatants were recovered. The Millex-GP 0.22 μm filter (Merck Millipore, Billerica, MA, USA) was used to filter the supernatants, then the filtrates were used as Stx preparations. 

In these preparations, the titers of Stx were determined using the VTEC-RPLA Seiken test (Denka Seiken Co., Ltd., Tokyo, Japan), following the manufacturer’s instructions. The titers of Stx1 and Stx2 were determined to be 256 and <2 in the Stx1, and <2 and 1024 in the Stx2 preparations, and <2 and <2 in the preparation without Stx, respectively. Titer of 128 was obtained for both purified Stx1 and Stx2 at 100 ng/mL and the concentrations of Stx1 and Stx2 were calculated according to the manufacturer’s instructions.

### 4.3. Determination of Stx-inhibitory Activity of Baicalein

The effects of baicalein (NAMIKI SHOJI Co., Ltd., Tokyo, Japan) on the cytotoxicity of Stx were determined according to the previously reported method [16]. Vero cells were purchased from Cell Bank (RIKEN BioResource Center, Tsukuba, Ibaraki, Japan). The cell viability was determined using the MTT Cell Proliferation Assay Kit (Cayman Chemical Company, Ann Arbor, MI, USA), following the supplier’s instruction. Medium was removed and fresh 5% FBS-MEM-E medium (10 µL) was added to the well. An amount of 12 mmol/L MTT stock solution (10 µL) was added to each well. The medium was removed after incubation at 37 °C. To resolve the MTT formazan crystals formed, 100 µL of Cell-Based Assay Buffer was added to the well. Medium alone was used as a negative control. Absorbance at 595 nm was measured using Microplate Reader Model 680 (Bio-Rad Laboratories Inc., Hercules, CA, USA). A value OD at 595 nm was used to show cell viability. 

To determine the inhibitory activity of baicalein, various concentrations of the Stx preparations were incubated with baicalein before addition to the Vero cell culture. The final concentration of baicalein in the culture was 0.027 and 0.13 mmol/L. Three replicates of the treatments were carried out per experiment and the values were shown as an average ± SD of A_595_. The statistical significance of the OD values was calculated by the Student’s *t*-test.

### 4.4. Determination of Protective Effect of Baicalein on Vero cells against Stx

The protective effects of baicalein on Vero cells against Stx were determined. Vero cell culture (0.1 mL) was added to each well of 96-well microtiter plate at 2 × 10^4^ cells per well and cultured for 24 h. One hundred microliters of baicalein at 0.08 and 0.4 mmol/L in PBS were added to each well and incubated for 1 h at 37 °C. Stx preparations (0.1 mL) were added to each well, then further cultured for 48 h at 37 °C in 5% CO_2_ incubator. The final concentrations of baicalein in the culture were 0.027 and 0.13 mmol/L. For control, PBS was used instead of the polyphenol. The viability of the Vero cells was determined using the MTT Cell Proliferation Assay Kit (Cayman Chemical Company, Ann Arbor, MI, USA), according to the supplier’s instruction.

Three replicates of the treatments were carried out per experiment. The values were reported as an average ± SD of A_595_. The statistical significance of the OD values was calculated by the Student’s T-test.

### 4.5. Determination of Transcription of Stx, Production and Secretion of StxF

To determine the effects of baicalein on transcription, production, and secretion of *stx*, *E. coli* O157:H7 No. 33 (*stx1*+, *stx2*−) and O157:H7 No. 36 (*stx1*−, *stx2*+) were used. These bacteria were inoculated into LB and cultured overnight at 37 °C with shaking. This bacterial culture was diluted in sterilized water to attain OD_660_ = 0.1 (bacterial concentration of ca. 10^8^ CFU per mL). The culture (1 mL) was centrifuged at 60,000× *g* for 5 min at 25 °C and the supernatants were recovered. The precipitate was resuspended in 1 mL sterile water and then 10-dilutied to a final concentration of ca. 10^6^ CFU/mL for use in subsequent experiments. One mL of 0.8 mmol/L baicalein solution was mixed with 1 mL of 2× LB and 100 µL of each of the diluted cultures in a test tube and the mixture was incubated at 37 °C with shaking at 130 rpm. The final concentration of baicalein in the mixture was 0.38 mmol/L. For control, sterile water was used instead of baicalein solution. To know the effects of Mitomycin C (MMC, Wao Pure Chemicals, Inc., Tokyo, Japan), LB (2 mL) and each of the diluted cultures (100 µL) were mixed and then incubated at 37 °C with shaking to reach OD = 0.1 before adding with MMC (final concentration of 0.2 mg/mL). To determine the effect of baicalein and MMC on transcription of *stx*, cells were harvested at OD_660_ ≈ 0.6. To determine the effects of baicalein on the production and secretion of Stx, the mixtures were incubated for 24 h.

To determine the effects of baicalein on transcription, production, and secretion of *stx*, *E. coli* O157:H7 No. 33 (*stx1*+, *stx2*−) and O157:H7 No. 36 (*stx1*−, *stx2*+) were used. These bacteria were inoculated into LB and cultured overnight at 37 °C with shaking. This bacterial culture was diluted in sterilized water to attain OD_660_ = 0.1 (bacterial concentration of ca. 10^8^ CFU per mL). The culture (1 mL) was centrifuged at 60,000× *g* for 5 min at 25 °C and the supernatants were recovered. The precipitate was resuspended in 1 mL sterile water and then 10-dilutied to a final concentration of ca. 10^6^ CFU/mL for use in subsequent experiments. One mL of 0.8 mmol/L baicalein solution was mixed with 1 mL of 2× LB and 100 µL of each of the diluted cultures in a test tube and the mixture was incubated at 37 °C with shaking at 130 rpm. The final concentration of baicalein in the mixture was 0.38 mmol/L. For control, sterile water was used instead of baicalein solution. To know the effects of Mitomycin C (MMC, Wao Pure Chemicals, Inc., Tokyo, Japan), LB (2 mL) and each of the diluted cultures (100 µL) were mixed and then incubated at 37 °C with shaking to reach OD = 0.1 before adding with MMC (final concentration of 0.2 mg/mL). To determine the effect of baicalein and MMC on transcription of *stx*, cells were harvested at OD_660_ ≈ 0.6. To determine the effects of baicalein on the production and secretion of Stx, the mixtures were incubated for 24 h.

To determine the amounts of transcript of *stx*, cells were harvested by centrifugation at 8000× *g* for 5 min at 4 °C. The precipitates were suspended in sterile water and the suspension was centrifuged at 8000× *g* for 5 min at 4 °C and the supernatants were removed. This process was repeated three times to wash the cells. A mixture of 700 µL Dw-saturated phenol and 700 µL TES buffer pre-warmed at 65 °C was added to the cell precipitate. After vortexing, the cell suspension was incubated for 40 min at 65 °C with occasional vortexing. The cell suspension was then placed on ice for 2 min and centrifuged at 17,860× *g* for 10 min at 4 °C. Eight hundred µL of TRIzol^®^ LS was added to the aqueous phase, and the mixture was vortexed and placed on ice for 5 min. After 200 µL of chloroform was added to the mixture, the mixture was vortexed, placed on ice for 5 min, and centrifuged at 17,860× *g* for 10 min at 4 °C. To the aqueous phase recovered, 600 µL of chloroform was added, and the mixture was vortexed, placed on ice for 5 min, and centrifuged at 17,860× *g* for 10 min at 4 °C. Six hundred µL of isopropanol was added to the aqueous phase recovered and the mixture was vortexed, placed on ice for 10 min, and centrifuged at 17,860× *g* for 10 min at 4 °C. The precipitate was dissolved in 200 µL of DEPC- treated deionized water. Six hundred µL of 99% ethanol and 20 µL of 3 M NaOAc (pH 5.2) were added and the mixture was stored at −80 °C for 60 min. After centrifugation (17,860× *g*, 15 min, 4 °C), the precipitate was dried at room temperature for 10 min and dissolved in 20 µL of DEPC-treated deionized water. Quality and quantity of total RNA were checked with a spectrophotometer. The RNA samples were treated with RNase-Free DNase set (QIAGEN, Dusseldorf, Germany), purified and concentrated using the RNeasy Mini Kit (QIAGEN, Dusseldorf, Germany) according to the supplier’s instruction. The purified RNA samples were reverse-transcribed using the ReverTra Ace^®^ qPCR RT Master Mix (TOYOBO, Osaka, Japan) according to the supplier’s instruction. For *stx1*, primers Stx1-F-n2 (5’-gttgcgaaggaatttacc-3’) and Stx1-R-n2 (5’-gtctgtaatggagtacctattg-3’), and for *stx2*, Stx2-F-n2 (5’-cgacccaacaaagttatg-3’), Stx2-R-n2 (5’-gggtgtggttaataacag-3’) were designed on the basis of the nucleotide sequences of *stx* genes (accession number: CP017444.1, CP012802.1) For the housekeeping gene, primers rrsA-F (5’-aggccttcgggttgtaaagt-3’) and rrsA-R (cggggatttcacatctgact) were designed on the basis of the nucleotide sequence of *rrsA* gene (accession number: J01859.1) PCR mixture (20 μL) was consisted of 10 μL of THUNDERBIRD® SYBR qPCR Mix (TOYOBO), 0.04 μL of 50 × ROX reference dye, 0.6 μL of 10 µmol/L forward primer, 0.6 μL of 10 µM reaverse primer, 2 μL of 25 ng/μL cDNA, and 6.76 μL of RT-PCR grade water. Real-time PCR was conducted in the condition of initial denaturation at 95 °C for 60 s, 40 cycles of denaturation at 95 °C for 15 s, annealing at 58 °C for 15 s, and extension at 72 °C for 30 s, and 1 cycle of denaturation at 95 °C for 30 s, annealing at 58 °C for 30 s, and denaturation at 95 °C for 30 s. All PCR reactions were run on M×3000P^®^ Real-Time PCR System and results were analyzed by MxPro^TM^ Software version 3.00 (Stratagene, La Jolla, CA, USA). 

To determine the effect of baicalein on the production and secretion of Stx, the mixtures were centrifuged at 3300× *g* for 15 min at 4 °C after incubation for 24 h. The supernatants were filtered through a Millex-GP filter (0.45 µm) and the filtrate was used as an extracellular sample. The precipitate was washed once with PBS by centrifugation and suspended with 2 mL of PBS. The suspension was then centrifuged at 3300× *g* for 15 min at 4 °C. The precipitate was suspended with 2 mL of PBS. To the suspension, polymyxin B (Pfizer Japan Inc.) was added to the culture. The final concentration of polymyxin B was 5000 U/mL. The culture was incubated for 30 min at 37 °C with occasional vortex and centrifuged at 3300× *g* for 15 min at 4 °C. The supernatant was filtered by Millex-GP filter (0.45 µm) and used as an intracellular sample. Extracellular samples were used for Stx sample in case of *E. coli* O157:H7 treated with MMC.

The titers of Stx in the samples were determined by using the VTEC-RPLA Seiken test kit and the concentrations of Stx1 and Stx2 were calculated as described above. For cytotoxicity test of the samples, Vero cell culture (0.1 mL) was added to each well of 96-well microtiter plate at 2 × 10^4^ cells per well and cultured for 24 h. Extracellular and intracellular samples at a different mixing ratio in PBS (200 µL) were added to each well and cells were cultured for 48 h at 37 °C in 5% CO_2_ incubator. The MTT Cell Proliferation Assay Kit was used to determine the viability of the Vero cells as described above. Three replicates of the treatments were performed for each experiment and values were reported as average or average ± SD of A_595_. The statistical significance of the OD values was calculated using the Student’s *t*-test.

## Figures and Tables

**Figure 1 toxins-11-00505-f001:**
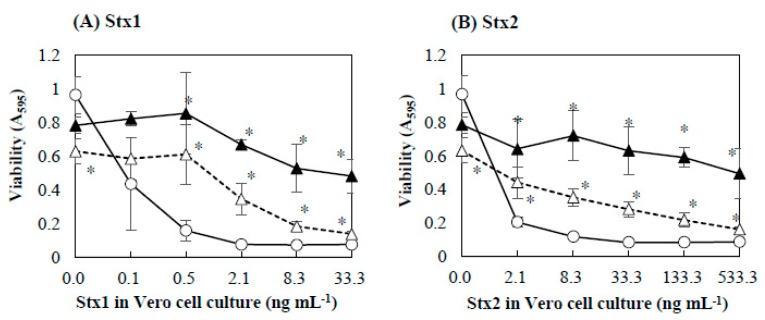
Effects of baicalein on the cytotoxicity of Stx1 and Stx2. Stx1 (**A**) and Stx2 (**B**) preparations respectively containing Stx1 and Stx2 were mixed with baicalein and incubated for 1 h at 37 °C. After the incubation, the mixture was added to the culture of Vero cells. The final concentrations of baicalein in the culture were 0 (○), 0.027 (△), and 0.13 (▲) mmol/L. Cell viability was determined by using MTT Cell Proliferation Assay after the cultivation at 37 °C for 48 h. Values are average ± SD for three separate experiments. *, *p* < 0.01.

**Figure 2 toxins-11-00505-f002:**
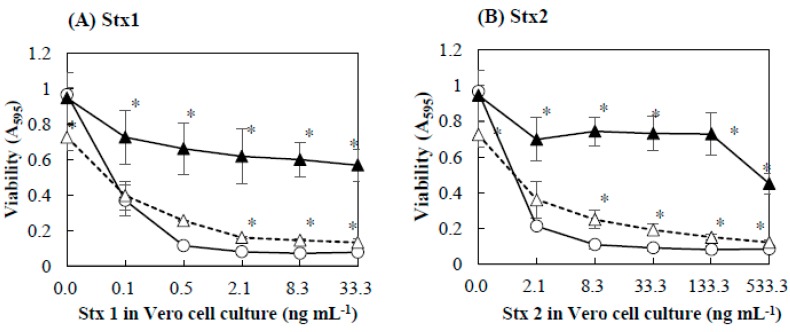
Effects of Stx1and Stx2 on viability of Vero cells pretreated with baicalein. Baicalein was added to Vero cell culture and incubated for 1 h at 37 °C. After the incubation, Stx1 (**A**) and Stx2 (**B**) preparations were added to the culture. The final concentrations of baicalein in the culture were 0 (○), 0.027 (△), and 0.13 (▲) mmol/L. After cultivation at 37 °C for 48 h, cell viability was determined by using MTT Cell Proliferation Assay. Values are average ± SD for three separate experiments. *, *p* < 0.01.

**Figure 3 toxins-11-00505-f003:**
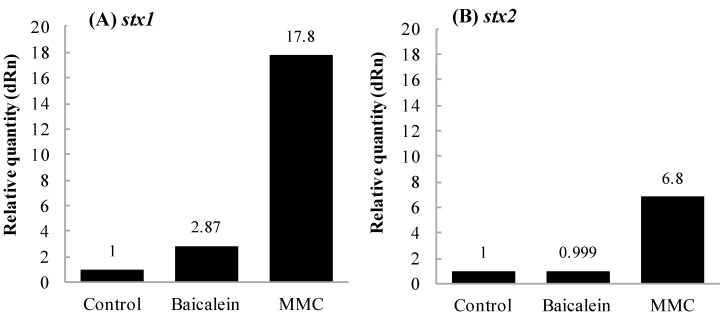
Effects of baicalein on the transcription of *stx* in *Escherichia coli* O157:H7. *E. coli* O157:H7 No. 33 (*stx1*+, *stx2*−) and O157:H7 No. 36 (*stx1*−, *stx2*+) strains were cultured until OD660 = 0.6 in the presence and absence of baicalein at 0.38 mmol/L. Amounts of transcripts of *stx1* (**A**) and *stx2* (**B**) were determined by real-time qPCR assay. Values are average of two separate experiments.

**Figure 4 toxins-11-00505-f004:**
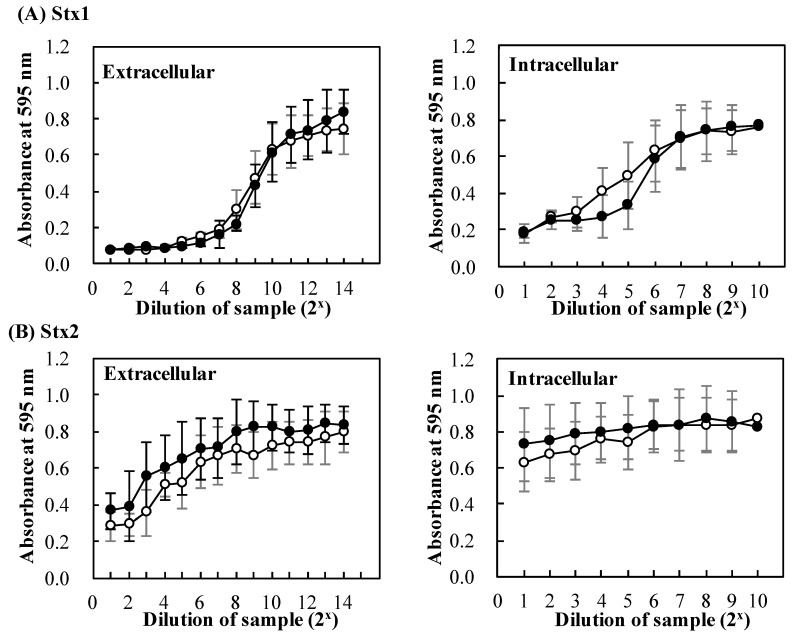
Effects of baicalein on secretion of Stx by *Escherichia. coli* O157:H7. *E. coli* O157:H7 No. 33 (*stx1*+, *stx2*−) and O157:H7 No. 36 (*stx1*−, *stx2*+) cultures were cultured in the presence (●) and absence (○) of baicalein for 24 h at 37 °C. After the incubation, extracellular and intracellular Stx1 (**A**), extracellular and intracellular Stx2 (**B**) preparations were respectively prepared from the cultures. The cytotoxicity of the preparations was determined on Vero cells. Cell viability was determined by using MTT Cell Proliferation Assay. The values are the average of three separate experiments.

**Figure 5 toxins-11-00505-f005:**
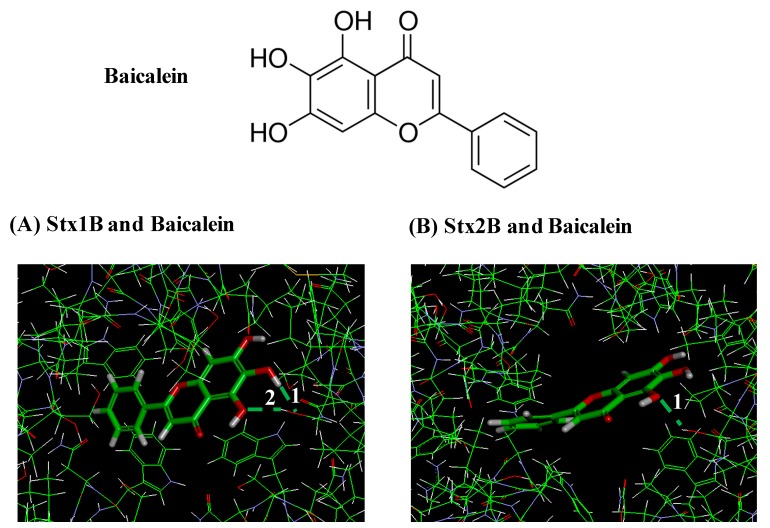
Docking models of baicalein bound to the pockets of Stx1B and Stx2B pentamers. Hydrogen bonds formed between baicalein and amino acids facing inside the pockets of Stx1B and Stx2B pentamers are white numbers. (**A**) Model showing two hydrogen bonds formed between baicalein and side chains of amino acid facing inside the pocket of the Stx1B pentamer at Ser42 and Ser42 of Monomer B. (**B**) Model showing one hydrogen bond formed between baicalein and side chain of amino acid of the Stx2B pentamer at Ser41 of Monomer J.

**Table 1 toxins-11-00505-t001:** Candidate compounds for Stx inhibitors selected from a collection of purified natural products isolated from plants (MEGxp; AnalytiCon Discover) by docking simulation.

ID	Name	MW	Interaction with Stx2B Pentamers Obtained by the Docking Model		Structure
			No. of Hydrogen Bond	Intramolecular Energy (cal/mol)	
NP-000438	Rhododendrol	166.22	3	<2	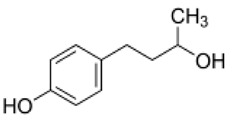
NP-000814	Acerogenin G	298.38	3	<1	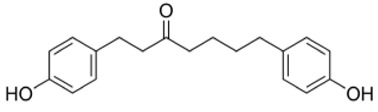
NP-001362	Esculetin	178.14	4	<2	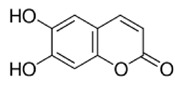
NP-003423	N-caffeoyltyramine	299.32	4	<2	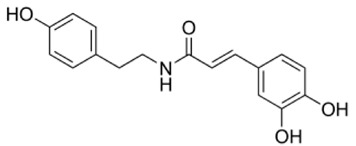
NP-003587	Thymol	150.22	0	<2	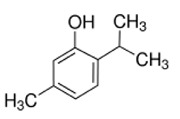
NP-003729	Baicalein	270.24	2	<1	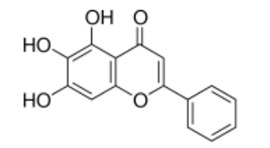
NP-003855	[6]-Gingerol	294.39	2	<1	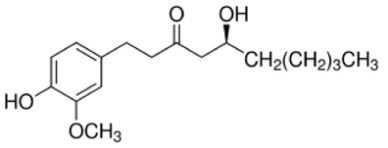

**Table 2 toxins-11-00505-t002:** Effects of baicalein on tx production after 24-h incubation.

Samples		Stx Concentration (ng/mL) **		
		Control	Baicalein	Mitomycin C
Stx1	Extracellular	400	400	1600
	Intracellular	50	50	ND
Stx2	Extracellular	200	200	204800
	Intracellular	3	3	ND

**: Stx concentration was determined by RPLA assay. Values are average of two separate experiments. ND: not determined.

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
