# Peer review of "Baicalein Inhibits Stx1 and 2 of EHE: Effects of Baicalein on the Cytotoxicity, Production, and Secretion of Shiga Toxins of Enterohaemorrhagic Escherichia coli"

_toxins, 2019, doi:10.3390/toxins11090505_

Round 1
Reviewer 1 Report
General comments:
In “Baicalein inhibits Stx1 and Stx2 of EHEC – effects of baicalein on the cytotoxicity, production and secretion of Shiga toxins of enterohaemorrhagic Escherichia coli” (toxins-569419), the investigators show that preincubation of Stx1 and Stx2 with two doses (0.027 and 0.13 mmol/L) of the plant flavone baicalein protects Vero cells from toxin-induced cytotoxicity at one time point (48 h) in vitro. Furthermore, the pretreatment of Vero cells with baicalein also protects the cells from cytotoxicity. Baicalein modestly increases the transcription of the gene encoding Stx1, but not Stx2. Baicalein treatment did not affect the production of Stx1 or Stx2 proteins found in extracellular or intracellular bacterial preparations. Toxin activities in extracellular and intracellular preparations prepared from EHEC cultured in the presence or absence of baicalein were not different (the assumption being that baicalein was removed from the preparations prior to addition to Vero cells). A docking model of baicalein bound to Stx1 and Stx2 B pentamers is presented although its relationship with toxin binding to the toxin receptor Gb3 is not discussed. The manuscript represents an initial attempt to define the interaction of a potential therapeutic plant product with Shiga toxins. There is very little discussion regarding the feasibility of using baicalein as a treatment option for hemorrhagic colitis or HUS. For example, are the concentrations used in these in vitro experiments attainable in vivo? How would baicalein be administered and would it be useful in patients already showing signs and symptoms of prodromal diarrhea or acute renal failure? The reviewer will not list individual grammatical errors, but English language usage is an issue in this manuscript. The authors are encouraged to seek editorial assistance to improve the paper. The reviewer offers ten specific comments for the authors’ consideration.
Specific comments:
1. Lines 27-28: It would be more accurate to say that Shiga toxin receptors are glycolipids of the globo-series, of which globotriaosylceramide (Gb3) is the primary receptor. Shiga toxins have been shown to also bind to Gb4, Gb5, and Forssman antigen. Gb3 expression is not limited to kidney epithelial cells. See Lee et al., Toxins 11:212 (2019) for a recent review of Shiga toxin binding to glycolipid receptors.
2) Lines 29-30: The inhibition of protein synthesis may be one mechanism by which Shiga toxins mediate cytotoxicity. However, a number of studies have shown that Shiga toxins induce apoptosis in some cell types. See Lee et al., Toxins 8:77 (2016) for a recent discussion on apoptosis induction by Shiga toxins.
3) Lines 32-33: This sentence does not make sense. The use of antibiotics to treat EHEC infections is controversial because the Shiga toxin operons are phage-encoded, and may be up-regulated by antibiotics that trigger an SOS response and phage lysis. Thus, the optimal expression of stx genes may require induction of the lambdoid phage lytic cycle.
4) Lines 61-64 and Table 1: In contrast to what is stated in the text, there is no information presented in Table 1 indicating decreases in Stx1 or Stx2 cytotoxicity after preincubation with the compounds. Rather, the data simply shows interactions of the compounds with the Stx2B pentamer. In line 65, please denote the meaning of MEGxp.
5) Lines 68-74 and Figure 1: What accounts for the consistent differences in Vero cell viability at the zero time point (1.0 vs. 0.8 vs. 0.6)? Does the left-most asterick in panels A and B indicate a significant difference (p < 0.01) at the zero time point? Did the investigators perform a baicalein only cytotoxicity control? The results shown are limited to single treatment times, that is, 1 hour preincubation with baicalein and 48 hours incubation with Vero cells. How do the values change when these time points are varied? Most importantly, if baicalein is administered after toxin is added to Vero cells, can the cells be
rescued from cell death? This experimental protocol would more likely reproduce a therapeutic approach to treating diarrhea or acute renal failure post-ingestion of EHEC.
6) Lines 82-88 and Figure 2: How do the investigators interpret the differences (or lack of differences) between the data shown in Figures 1 and 2. The major difference in the experimental protocols seems to be the time of exposure of toxins to baicalein; one hour preincubation vs. no preincubation. If in fact baicalein completes with Gb3 for toxin binding, then one might predict that in the experiments involving no preincubation with baicalein, the toxin-binding glycolipid on the surface of Vero cells might better compete with baicalein for toxin binding, resulting in greater cytotoxicity.
7) Lines 96-100 and Figure 3: What was the rationale for performing these experiments? Is there evidence in the literature that baicalein regulates gene expression in bacteria? Are E. coli O157:H7 strains 33 and 36 isogenic except for toxin production?
8) Lines 114-119 and Figure 4: Additional experimental details are necessary to interpret the data shown in Figure 4, especially the statement “There was no significant difference in viability between Vero cells in the presence of Stx prepared from E. coli O157:H7 cultured in presence and absence of baicalein.” According to the data in Figures 1 and 2, this statement should not be true. The reviewer had to read the Materials and Methods section carefully to learn that the investigators used filtered supernatants in these assays, so that baicalein would have been removed from the preparations. A brief description of how these experiments were performed would save the reader confusion about the role (or lack thereof) on baicalein on cytotoxicity. There is no negative control (extracellular and intracellular filtered supernatants prepared from non-toxigenic E. coli) shown for these experiments.
9) Lines 128-140 and Figure 5: How do the docking models of baicalein to Stx1B and Stx2B pentamers correlate with current understanding of B pentamer interaction with Gb3? For example, structure/function studies have suggested that each B monomer may possess 2 or 3 Gb3 binding sites so that the pentamer is capable of binding 10-15 Gb3 sites. Do the data shown in Figure 5 support the concept that baicalein physically blocks the interaction of the toxin molecule with toxin receptor?
10) Lines 178-180: Where in this paper are the data to support the statement: “It seems that baicalein protects Vero cells from cytotoxicity of Stx by binding to the surface of the cytoplasmic membrane of the cell and altering the function of the membrane.” The inclusion of data supporting this statement would greatly strengthen the study. The investigators have shown evidence that baicalein binds to Shiga toxin B-pentamers, but not to Vero cell cytoplasmic membranes.
Author Response
Dear reviwer,
Thank you for your kind reviewing our article (Manuscript ID: toxins-569419) entitled ‘Effect of baicalein on the cytotoxicity, production and secretion of Shiga toxins of enterohaemorrhagic Escherichia coli”. According to your comment, we revised the manuscript and made response sheet to answer to each comment from you. If our revisions are still inadequate, your further kind suggestions would be appreciated for improving our revised manuscript.
Comment from you:
General comments:
In “Baicalein inhibits Stx1 and Stx2 of EHEC – effects of baicalein on the cytotoxicity, production and secretion of Shiga toxins of enterohaemorrhagic Escherichia coli” (toxins-569419), the investigators show that preincubation of Stx1 and Stx2 with two doses (0.027 and 0.13 mmol/L) of the plant flavone baicalein protects Vero cells from toxin-induced cytotoxicity at one time point (48 h) in vitro. Furthermore, the pretreatment of Vero cells with baicalein also protects the cells from cytotoxicity. Baicalein modestly increases the transcription of the gene encoding Stx1, but not Stx2. Baicalein treatment did not affect the production of Stx1 or Stx2 proteins found in extracellular or intracellular bacterial preparations. Toxin activities in extracellular and intracellular preparations prepared from EHEC cultured in the presence or absence of baicalein were not different (the assumption being that baicalein was removed from the preparations prior to addition to Vero cells). A docking model of baicalein bound to Stx1 and Stx2 B pentamers is presented although its relationship with toxin binding to the toxin receptor Gb3 is not discussed. The manuscript represents an initial attempt to define the interaction of a potential therapeutic plant product with Shiga toxins. There is very little discussion regarding the feasibility of using baicalein as a treatment option for hemorrhagic colitis or HUS. For example, are the concentrations used in these in vitro experiments attainable in vivo? How would baicalein be administered and would it be useful in patients already showing signs and symptoms of prodromal diarrhea or acute renal failure? The reviewer will not list individual grammatical errors, but English language usage is an issue in this manuscript. The authors are encouraged to seek editorial assistance to improve the paper. The reviewer offers ten specific comments for the authors’ consideration.
Answer: Thank you for your careful reviewing. We check the manuscript again and also asked native speaking colleagues to check it. If our revised manuscript is not enough to be accepted, could you please let us know?
Specific comments:
Lines 27-28: It would be more accurate to say that Shiga toxin receptors are glycolipids of the globo-series, of which globotriaosylceramide (Gb3) is the primary receptor. Shiga toxins have been shown to also bind to Gb4, Gb5, and Forssman antigen. Gb3 expression is not limited to kidney epithelial cells. See Lee et al., Toxins 11:212 (2019) for a recent review of Shiga toxin binding to glycolipid receptors.
Answer: We really appreciate your kind reviewing. According to your comments, we modified the manuscript as “The Stx receptors are glycolipids of the globo-series, in which globotriaosylceramides (Gb3s) being the primary receptor is found on the surface of kidney epithelial cells. The Stx B subunit interacts with Gb3 and induces membrane invagination, leading to the internalization of the toxin. The internalized Stx toxin inhibits protein synthesis, leading to cell death. Recently, however, Lee et al. (2016) have also presented on the multifuncationality of Stxs that not only inhibits protein synthesis but aslo induced apoptosis in different cell types [3]. In addition, Stx variants have been shown to also bind to different receptors. Stx1a preferential binding to Gb3 and only detectably binds with Gb4 whereas Stx2a strongly binds to Gb3 and marginally bind to globotetraosylceramide (Gb4) [4]. Beside, Stx2e binds to both Gb3 and Gb4, also to the Forssman glycosphingolipid [5].” (P.1. Lines 27-35)
2) Lines 29-30: The inhibition of protein synthesis may be one mechanism by which Shiga toxins mediate cytotoxicity. However, a number of studies have shown that Shiga toxins induce apoptosis in some cell types. See Lee et al., Toxins 8:77 (2016) for a recent discussion on apoptosis induction by Shiga toxins.
Answer: Thank you for your important suggestion. In the manuscript, we mentioned “Baicalein has also been reported to inhibit the biofilm formation of Candida albicans [19], and to induce apoptosis in a variety of human cancer cell lines [20–23].” (Line 1551-153). In revised manuscript, we added this information.
Recently, however, Lee et al. (2016) have also presented on the multifuncationality of Stxs that not only inhibits protein synthesis but aslo induced apoptosis in different cell types [3]. (P.1; Lines 30-32)
3) Lines 32-33: This sentence does not make sense. The use of antibiotics to treat EHEC infections is controversial because the Shiga toxin operons are phage-encoded, and may be up-regulated by antibiotics that trigger an SOS response and phage lysis. Thus, the optimal expression of stx genes may require induction of the lambdoid phage lytic cycle.
Answer: Thank you for your careful reviewing. We rewrote the sentence in lines 32-33 as below: However, antibiotic therapy in treament EHEC infection is still controversial because the increas of Stx production and secretion lead to the risk of HUS development [6–8]. (P.1; Lines 36-38)
As for the discussion, in the manuscript, we mentioned some reasons lead to the increase of Stx production to explain for the effects of baicalein on transcription of sxt in Escherichia coli O157:H7 (P.7; Line 188-195)
If our information still has problems, could you please let us know?
4) Lines 61-64 and Table 1: In contrast to what is stated in the text, there is no information presented in Table 1 indicating decreases in Stx1 or Stx2 cytotoxicity after preincubation with the compounds. Rather, the data simply shows interactions of the compounds with the Stx2B pentamer. In line 65, please denote the meaning of MEGxp.
Answer: Thank you for your comment. We are sorry for our insufficient information about the cytotoxicity of Stx1 and Stx2 after preincubation with compounds, we attached the profiles of the cytotoxicity of Stx1 and Stx2 after preincubation with compounds as the supplementary data (Figure S-1), as below:
Supplementary Figure S-1. Effects of polyphenols on the cytotoxicity of Stx1 and Stx2. Stx1 (A) and Stx2 (B) preparations containing Stx1 and Stx2 at 12.5 and 50 mg/L, respectively were blended without (None) or with 100 mg/L of each of the polyphenols and incubated at 37oC for 1 h. After the incubation, the mergence was added to the culture of Vero cells. Cell viability was determined by using MTT Cell Proliferation. Assay after the cultivation at 37 C for 24. Values are average of two separate experiments.
As your suggestion, we added denote the meaning of MEGxp as follows: MEGxp (AnalytiCon Discover) (P.2; Line 70-71)
5) Line 68-74 and Figure 1: What accounts for the consistent differences in Vero cell viability at the zero time point (1.0 vs. 0.8 vs. 0.6)? Does the left-most asterick in panels A and B indicate a significant difference (p < 0.01) at the zero time point? Did the investigators perform a baicalein only cytotoxicity control? The results shown are limited to single treatment times, that is, 1 hour preincubation with baicalein and 48 hours incubation with Vero cells. How do the values change when these time points are varied? Most importantly, if baicalein is administered after toxin is added to Vero cells, can the cells be rescued from cell death? This experimental protocol would more likely reproduce a therapeutic approach to treating diarrhea or acute renal failure post-ingestion of EHEC. How do the values change when these time points are varied? Most importantly, if baicalein is administered after toxin is added to Vero cells, can the cells be rescued from cell death? This experimental protocol would more likely reproduce a therapeutic approach to treating diarrhea or acute renal failure post-ingestion of EHEC.
Answer: As you mentioned, there are consistent differences in Vero cell viability at the concentration Stxs 0 ng/mL because the cytotoxicity of baicalein effects on Vero cell. Accroding to Zandi et al. 2012 reported that the viability of Vero cell reduce by 50% by the concentration of baicalein at 109 μg/mL. However, baicalein in particular and flavonoids in general were described as less toxic in comparison to other plant compounds (alkaloids). In this study, we would like focus on the effects of baicalein on the cytotoxicity of Stx1 and Stx2 instead of inhibition of baicalein on Vero cell growth.
We again like to claim the prominent subjects on this study to elucidate: 1. Baicalein strongly reduced the cytotoxicity of both Stx1 and Stx2; 2. Baicalein protected Vero cells from the cytotoxicity of both Stx1 and Stx2; 3. Baicalein increased transcription of stx1 but not of stx2. However, baicalein had no effects on the production and secretion of Stxs, though it reduced cytotoxicity of Stx and protected Vero cells at the lower concentration; 4. Baicalein seems to form a stable hydrogen bond with the side chain of amino acid facing inside the pocket of the Stx2B pentamer.
We performed the experiment for 1-hour pretreatment with baicalein to evaluate the sensitivity of baiclein to the cytotoxicity and 48-hours incubation with Vero cell yielded optimal synergistic cytotoxicity of Stx to Vero cell.
Thank you for critical comments. As you mentioned, if baicalein is administered after toxin is added to Vero cells, can the cells be rescued from cell death? This is an important method to therapeutic approach to treating diarrhea or acute renal failure post-ingestion of EHEC. It is really regrettbale that we were not able to done that in this study. Again, we would like to thank you for your comment, your suggestion may be a promising approach to further researchs.
6) Lines 82-88 and Figure 2: How do the investigators interpret the differences (or lack of differences) between the data shown in Figures 1 and 2. The major difference in the experimental protocols seems to be the time of exposure of toxins to baicalein; one hour preincubation vs. no preincubation. If in fact baicalein completes with Gb3 for toxin binding, then one might predict that in the experiments involving no preincubation with baicalein, the toxin-binding glycolipid on the surface of Vero cells might better compete with baicalein for toxin binding, resulting in greater cytotoxicity.
Answer: Thank you for your suggestion. As you mentioned, there is difference the data shown Figure 1 and 2 migh be due to difference in the experimental protocols lead to between binding of toxin – baicalein, toxin – the suface of Vero cell. We added the explanation for the issue to the revised the manuscript.
The inhibition effects of baicalein on the cytotoxicity of Stx were determined (Figure 1 and Figure 2). However, as the results showed in Figure 1 and 2, there was a decrease in inhibition of bacalein at 0.027 mmol/L in Figure 2 compared to the results in Figure 1. As we above mentioned, baicalein inhibits the cytotoxicity of both Stx1 and Stx2, possibly by forming a stable binding structure at the pocket of the Stx1B and Stx2B pentamers. In a recent report of Cherubin et al. (2016) have demonstrated that some polyphenols inhibit Cholera toxin, an AB-5 toxin member produced by Vibrio cholerae, by preventing the binding of cholera toxin of B subunit to its GM1-gangelioside receptor [38]. It suggested that the binding of baicalein to StxB pentamers lead to preventing the binding of Stx to glycolipid on the surface of Vero cell. In this study, the inhibiton effect of baicalein decrease since the binding of Stx to glycolipid on the surface of Vero cell was more preferential than the binding of baicalein to StxB pentamers. (P7. Lines 202-212)
7) Lines 96-100 and Figure 3: What was the rationale for performing these experiments? Is there evidence in the literature that baicalein regulates gene expression in bacteria? Are E. coli O157:H7 strains 33 and 36 isogenic except for toxin production?
Answer: Thank you for your comment. As we mentioned the use of antibiotics to treat EHEC infections is controversial because the increas of Stx production and secretion lead to the risk of HUS development. So, we performed the experiments to determine the effects of baicalein on transcirption of stx1 and stx2 in compared with negative control (water) and positive control (Mitomycin C – an antibiotic was used to treat bacteria infection).
As you mentioned, Is there evidence in the literature that baicalein regulates gene expression in bacteria? According to Cai et al., 2016 have reported that baicalein significantly inhibited the expression of CTX-M-1 mRNA expression in Klebsiella pneumoniae strains.
Some E.coli O157:H7 strains produce only one toxin type, etheir Stx1 or Stx2 while others express both Stx1 and Stx2. For each strain: E.coli O157:H7 No.33 (stx1+, stx2-) and E.coli O157:H7 No.36 (stx1-, stx2+) (+) denotes E.coli produce toxin type, (-) denotes E.coli strain do not produce toxin. Sub-type of Stx1 strain 33 was Stx1a, sub-type of Stx2 strain 36 and 184 were Stx2a. As we mentioned as above, these strains isolated by our laboratory and the results have been published
8) Lines 114-119 and Figure 4: Additional experimental details are necessary to interpret the data shown in Figure 4, especially the statement “There was no significant difference in viability between Vero cells in the presence of Stx prepared from E. coli O157:H7 cultured in presence and absence of baicalein.” According to the data in Figures 1 and 2, this statement should not be true. The reviewer had to read the Materials and Methods section carefully to learn that the investigators used filtered supernatants in these assays, so that baicalein would have been removed from the preparations. A brief description of how these experiments were performed would save the reader confusion about the role (or lack thereof) on baicalein on cytotoxicity. There is no negative control (extracellular and intracellular filtered supernatants prepared from non-toxigenic E. coli) shown for these experiments.
Answer: Thank you for your important comments and we are sorry for our lack of performing these experiment. In Figure 4, we would like to evaluated the effects of baicalein on secretion of Stx by E.coli in detail. Baicalein was added in E.coli culture and incubated 37oC to attain. After incubation, extracellular and intracellular Stxs were preparated from these cultures and then extracellular and intracellular Stxs preparations were added in Vero cells, incubated at 37oC for 48h. The results showed baicalein had no effect on the secretion of Stx by E.coli.
For extracellular and intracellular Stxs preparations, after incubation, these cultures were centrifuged at 3,300×g for 15 min at 4°C. The supernatants were filtered through a Millex-GP filter (0.45 μm). This filter had no effect on the production and secretion of Stx by E.coli strains. Hence, it had no effect on the inhibition effect of baicalein agiants Stxs.
According to data shown in Figure 1, Stxs were preincubated with baicalein at 37oC for 1h before add to Vero cell culture incubated 37oC for 48h. The results showed the viability of Vero cells in the presence of Stx1 or Stx2 treated with baicalein was significantly higher than untreated control. In Figure 2, Stxs were added to Vero cell pretreated with baicalein. As regard to Figure 2, in the presence of Stx1 or Stx2, the viability of Vero cells treated with baicalein was significantly higher than untreated control. Hence, there was no conflict among these figures.
We are sorry, in this study, we did not perform a negative control (extracellular and intracellular filtered supernatants prepared from non-toxigenic E. coli) in our experiments.
9) Lines 128-140 and Figure 5: How do the docking models of baicalein to Stx1B and Stx2B pentamers correlate with current understanding of B pentamer interaction with Gb3? For example, structure/function studies have suggested that each B monomer may possess 2 or 3 Gb3 binding sites so that the pentamer is capable of binding 10-15 Gb3 sites. Do the data shown in Figure 5 support the concept that baicalein physically blocks the interaction of the toxin molecule with toxin receptor?
Answer: Thank you for critical comments. We added the discussion to the revised manuscript and answer for your comment 6.
If our revised manuscript is not enough to be accepted, could you please let us know?
10) Lines 178-180: Where in this paper are the data to support the statement: “It seems that baicalein protects Vero cells from cytotoxicity of Stx by binding to the surface of the cytoplasmic membrane of the cell and altering the function of the membrane.” The inclusion of data supporting this statement would greatly strengthen the study. The investigators have shown evidence that baicalein binds to Shiga toxin B-pentamers, but not to Vero cell cytoplasmic membranes
Respond: Thank you for your comment. In this paper, we had no data to support the statement “It seems that baicalein protects Vero cells from cytotoxicity of Stx by binding to the surface of the cytoplasmic membrane of the cell and altering the function of the membrane”.
However, we based on the relevant publications to explain for this statement. “In addition, Cariddi et al. (2015) reported that caffeic acid, a polyphenol, protected Vero cells from the cytotoxicity of ochratoxin A by altering principally the lysosomal function in Vero cells [33]. It seems that baicalein protects Vero cells from cytotoxicity of Stx by binding to the surface of the cytoplasmic membrane of the cell and altering the function of the membrane.”. (P.7; Lines 190-194)
We appreciate your important suggestion to make our manuscript much more valuable.
If the revised manuscript is still insufficient, could you please give us further comments.
Reviewer 2 Report
Dear Editor,
This manuscript describes the “Effects of baicalein on the cytotoxicity, production and secretion of Shiga toxins of enterohaemorrhagic Escherichia coli.” They used a well-established Vero cell assay to demonstrate the reduction in cytotoxicity of Stx1 and Stx2 (subtypes not defined). They also used software to show potential sites in the pentameric B subunit complex where baicalein could bind. The concentration of baicalein required to exert a significant effect is on the order of 130 micromolar.
These results would be interesting had other researchers not previously demonstrated the efficacy of a glycosylated form of baicalein, baicalin.[1,2] Both baicalein and baicalin are isolated from a Chinese herb, Scutellaria baicalensis. This reviewer examined two papers, but there is a larger body of work. These other researchers used mutant forms of stx2 to map out the baicalin binding site. They determined that it induced the formation of oligomers by binding to the catalytically active A subunit.[1] Furthermore, the activity of baicalin is approximately 36 micromolar or about five-fold lower than baicalein.
The authors of the reviewed manuscript did not cite the work relating baicalin and Shiga toxin. They did cite work describing the anti-inflammatory potential of baicalin and its aglycone baicalein.[3] The authors need to consider the relevance of the baicalin/Shiga toxin work of others. The authors elected to model the docking of baicalein using a crystal structure that solely consisted of the pentameric B subunits (3MXG). It would be interesting to see what would happen it they used an intact Shiga toxin in their model.
The Shiga toxins produced by Shiga toxin producing Escherichia coli (STEC) are produced under the strict control of a temperate lambdoid phage (stx phage).[4] When the stx phage senses stress in the host E. coli (via the SOS response) they begin to replicate lytically which leads to the production of intact phages, Shiga toxins, and the death of the host E. coli. Some antibiotics such as penicillin, ciprofloxacin, and mitomycin C induce the SOS response, which leads to the excessive production of Shiga toxins. Antibiotics that inhibit protein synthesis (e. g. tetracycline, erythromycin, some amino glycosides) prevent the production of Shiga toxins. The only know exception is when a E. coli (infected with some Stx1 phages) host is grown under iron poor conditions. On a minor note some Shiga toxins bind to Gb3 and Gb4.[5]
The authors need to revise their introduction and discussion to appropriately incorporate the previous baicalin work and the stx phage control of Shiga toxin production.
Other comments:
The graphs in Figure 1 seem to show that Baicalein inhibits Vero cell growth. Adding 0, 0.027 or 0.13 mmol/L results in a basal Vero cell viability (395 nm) of 1, 0.6 and 0.8 respectively. The authors do not comment on this.
Why was 0.13 micromolar Baicalein used in the Vero cell assay and 0.38 micromolar Baicalein used in the bacterial cell assay.
In Figure 4 strains 33 and 36 were used. In the experimental section strains 33 and 184 were used. Was strain 184 or 36 or both used in this work? Which culture collection holds these strains? The experiments were done in the presence and absence of Baicalein. How much Baicalein was present 0.13 or 0.38 micromolar?
The authors do not state which sub-type of Stx1 strain 33 produces. They do not state which subtype of stx2 is produced by strain 36 or 184.
Line 133-134
(A-E and A-J were named of each the monomers of Stx1B and Stx2B pentamers, respectively) implies that the authors use a capital letter to designate type variants. The subtype variants are designated lower case letters (Stx1a-e and Stx2a-j)
They authors use the term Stx1B and Stx2B pentamers. They should define this as the pentameric B subunits of Stx1 and Stx2 to avoid confusion. Stx1b and Stx2b have different pentameric B subunits.
In the discussion the authors note the properties of Baicalein. They should not that it is the aglucone of Baicalin, a potent inhibitor of Shiga toxin.
Line 44 O57:H7 should be O157:H7
Lines 281-282. Did the authors spin at 60,000 x g? That seems to be excessive. The pellet and not the precipitate was resuspended.
coli O157:H7 No.33 (stx1+, stx2-) and O157:H7 No.184 (stx1-, stx2+) what do they produce?
Dong J, Zhang Y, Chen Y, Niu X, Zhang Y, Yang C, Wang Q, Li X, Deng X. 2015. Baicalin inhibits the lethality of Shiga-like toxin 2 in mice. Antimicrob Agents Chemother. 2015 Nov;59(11):7054-60. Zhang Y, Qi Z, Liu Y, He W, Yang C, Wang Q, Dong J, Deng X. 2017. Baicalin Protects Mice from Lethal Infection by Enterohemorrhagic Escherichia coli. Front Microbiol. 8:395. Dinda B, Dinda S, DasSharma S, Banik R, Chakraborty A, Dinda M. Therapeutic potentials of baicalin and its aglycone, baicalein against inflammatory disorders. Journal of Medicinal Chemistry 2017, 68–80. O'Brien AD, Newland JW, Miller SF, Holmes RK, Smith HW, Formal SB. 1984. Shiga-Like Toxin-Converting Phages from Eschenrchia coli Strains That Cause Hemorrhagic Colitis or Infantile Diarrhea. Science 226: 694-696. Karve SS, Weiss AA. 2014. Glycolipid binding preferences of Shiga toxin variants. PLoS One. 9(7): e101173.
Author Response
Dear reviwer,
Thank you for your kind reviewing our article (Manuscript ID: toxins-569419) entitled ‘Effect of baicalein on the cytotoxicity, production and secretion of Shiga toxins of enterohaemorrhagic Escherichia coli”. According to your comment, we revised the manuscript and made response sheet to answer to each comment from you. If our revisions are still inadequate, your further kind suggestions would be appreciated for improving our revised manuscript.
Comment from you:
This manuscript describes the “Effects of baicalein on the cytotoxicity, production and secretion of Shiga toxins of enterohaemorrhagic Escherichia coli.” They used a well-established Vero cell assay to demonstrate the reduction in cytotoxicity of Stx1 and Stx2 (subtypes not defined). They also used software to show potential sites in the pentameric B subunit complex where baicalein could bind. The concentration of baicalein required to exert a significant effect is on the order of 130 micromolar.
These results would be interesting had other researchers not previously demonstrated the efficacy of a glycosylated form of baicalein, baicalin.[1,2] Both baicalein and baicalin are isolated from a Chinese herb, Scutellaria baicalensis. This reviewer examined two papers, but there is a larger body of work. These other researchers used mutant forms of stx2 to map out the baicalin binding site. They determined that it induced the formation of oligomers by binding to the catalytically active A subunit.[1] Furthermore, the activity of baicalin is approximately 36 micromolar or about five-fold lower than baicalein.
The authors of the reviewed manuscript did not cite the work relating baicalin and Shiga toxin. They did cite work describing the anti-inflammatory potential of baicalin and its aglycone baicalein.[3] The authors need to consider the relevance of the baicalin/Shiga toxin work of others. The authors elected to model the docking of baicalein using a crystal structure that solely consisted of the pentameric B subunits (3MXG). It would be interesting to see what would happen it they used an intact Shiga toxin in their model.
The Shiga toxins produced by Shiga toxin producing Escherichia coli (STEC) are produced under the strict control of a temperate lambdoid phage (stx phage).[4] When the stx phage senses stress in the host E. coli (via the SOS response) they begin to replicate lytically which leads to the production of intact phages, Shiga toxins, and the death of the host E. coli. Some antibiotics such as penicillin, ciprofloxacin, and mitomycin C induce the SOS response, which leads to the excessive production of Shiga toxins. Antibiotics that inhibit protein synthesis (e. g. tetracycline, erythromycin, some amino glycosides) prevent the production of Shiga toxins. The only know exception is when a E. coli (infected with some Stx1 phages) host is grown under iron poor conditions. On a minor note some Shiga toxins bind to Gb3 and Gb4.[5]
The authors need to revise their introduction and discussion to appropriately incorporate the previous baicalin work and the stx phage control of Shiga toxin production.
Answer: Thank you for your careful reviewing. We appreciate your important suggestion to make our manuscript much more valuable. We checked the manuscript again and revised it.
If the revised manuscript is still insufficient, could you please give us further comments.
Other comments:
The graphs in Figure 1 seem to show that Baicalein inhibits Vero cell growth. Adding 0, 0.027 or 0.13 mmol/L results in a basal Vero cell viability (395 nm) of 1, 0.6 and 0.8 respectively. The authors do not comment on this.
Answer: Thank you for your reviewing and advice. As you mentioned, there are consistent differences in Vero cell viability at the concentration Stxs 0 ng/mL because the cytotoxicity of baicalein effects on Vero cell. Accroding to Zandi et al. 2012 reported that the viability of Vero cell reduce by 50% by the concentration of baicalein at 109 μg/mL. However, baicalein in particular and flavonoids in general were described as less toxic in comparison to other plant compounds (alkaloids). In this study, we would like focus on the effects of baicalein on the cytotoxicity of Stx1 and Stx2 instead of inhibition of baicalein on Vero cell growth.
Why was 0.13 micromolar Baicalein used in the Vero cell assay and 0.38 micromolar Baicalein used in the bacterial cell assay.
Answer: Thank you for your comment. To demonstrate the effect of baicalein on the cytotoxicity Stx1 and Stx2, Stx1 and Stx2 were mixed with baicalein and incubated for 1h at 37°C. After the incubation, the mixture was added to the culture of Vero cells. Therefore, baicalein directly effects on the cytotoxicity of Stx and the concentration 0.13 mmol/L of baicalein was used. However, to demonstrate the effects of baicalein on productivity of Stx by enterohaemorrhagic E.coli, baicalein was added E.coli culture and incubated for 24 h at 37oC, After the incubation, extracellular and intracellular Stx were prepared from this cultures, then extracellular and intracellular Stx were added to Vero cell culture. In this experiment, baicalein not only effects on growth of E.coli but also on Stx production and secretion by E.coli. Furthermore, the minimum inhibitory concentration (MIC) of baicalein against Klebsiella pneumoniae - a type of Gram-negative bacteria is 256 µg/mL (Cai et al., 2016) and that of baicalin to E.coli is 4000 μg/mL (Zhao et al., 2017). Hemce, the concentration 0.38 mmol/L of baicalein was used.
In Figure 4 strains 33 and 36 were used. In the experimental section strains 33 and 184 were used. Was strain 184 or 36 or both used in this work? Which culture collection holds these strains? The experiments were done in the presence and absence of Baicalein. How much Baicalein was present 0.13 or 0.38 micromolar?
Answer: Thank you for your reviews. In this study, we used both strains 184 and 36 which product only Stx2.
These strains were isolated from raw meat samples by our laboratory. These strains were Shiga toxin-producing E.coli and compared with those from ill humans. After isolation and identification, the strains were stored in tryptic soy broth containing 25% (v/v) glycerol at 80°C for further research. These results have been published by Hoang Minh et al., 2015.
The experiments were done in the presence and absence of baicalein. To demonstrate the effects of baicalein on the cytotoxicity Stx1 and Stx2 and the protective effect of baicalein on Vero cell against Stx1 and Stx2, the final concentration of baicalein was 0.13 mmol/L. To demonstrate the effects of baicalein on productivity of Stx by enterohaemorrhagic E.coli, the final concentration of baicalein was 0.38 mmol/L.
The authors do not state which sub-type of Stx1 strain 33 produces. They do not state which subtype of stx2 is produced by strain 36 or 184.
Answer: Thank you for your kind suggestion. For each strain: E.coli O157:H7 No.33 (stx1+, stx2-) and E.coli O157:H7 No.148 (stx1-, stx2+), O157:H7 No.148 (stx1-, stx2+). (+) denotes E.coli produce toxin type, (-) denotes E.coli strain do not produce toxin type.
Sub-type of Stx1 strain 33 was Stx1a, sub-type of Stx2 strain 36 and 184 were Stx2a. As we mentioned as above, these strains isolated by our laboratory and the results have been published.
Line 133-134
(A-E and A-J were named of each the monomers of Stx1B and Stx2B pentamers, respectively) implies that the authors use a capital letter to designate type variants. The subtype variants are designated lower case letters (Stx1a-e and Stx2a-j)
Answer: Thank you for your comment. As show in line 137 and 139, we mentioned the hydrogen bonds formed between baicalein and side chains of amino acids at Ser42 Monomer B of the Stx1B and Ser41 Monomer J of Stx2B. According to the report by Miyamoto et al., 2014, A-E and A-J were only named of each the monomers of Stx1B and Stx2B pentamers, respectively.
They authors use the term Stx1B and Stx2B pentamers. They should define this as the pentameric B subunits of Stx1 and Stx2 to avoid confusion. Stx1b and Stx2b have different pentameric B subunits.
Answer: Thank you for your comment. To shorten the characters, the pentameric B subunits of Stx types 1 are Stx1B and the pentameric B subunits of Stx types 2 are Stx2B.
In the discussion the authors note the properties of Baicalein. They should not that it is the aglucone of Baicalin, a potent inhibitor of Shiga toxin.
Answer: As your suggestion, we added a slight discussion for approaches in the revised manuscripts as below:
According Dong et al. (2015) reported that baicalin deriving to bacalein significantly reduced the activity of Stx2 which induced lethality in mice [29]. The protection effecs of baicalin on cell against EHEC infection was clarified by Zhang et al. (2017) [30]. (P.7. Lines: 166-168)
Line 44 O57:H7 should be O157:H7
Answer: We corrected “O57:H7” to “O157:H7” according to your suggestion. (P.2. Line 48)
Lines 281-282. Did the authors spin at 60,000 x g? That seems to be excessive. The pellet and not the precipitate was resuspended.
Answer: Thank you for your suggsetion. There were many studies, they performed the experiment to obtain concentrations of bacterial cultural supernatant by centrifugation at 60000 x g such as Weinstock et al., (1983); Allet et al. (1988) and Xie et al., (2017).
E.coli O157:H7 No.33 (stx1+, stx2-) and O157:H7 No.184 (stx1-, stx2+) what do they produce?
Answer: Thank you for your comment. Some E.coli O157:H7 strains produce only one toxin type, etheir Stx1 or Stx2 while others express both Stx1 and Stx2. For each strain: E.coli O157:H7 No.33 (stx1+, stx2-) and E.coli O157:H7 No.148 (stx1-, stx2+); (+) denotes E.coli produce toxin type, (-) denotes E.coli strain do not produce toxin.
Allet, B., Payton, M., Mattaliano, R.J., Gronenborn, A.M., Clore, G.M. and Wingfield, P.T. (1988) Purification and characterization of the DNA-binding protein Ner of bacteriophage Mu. Gene. 65, 259–268.
Weinstock, G.M., ap Rhys, C., Berman, M.L., Hampar, B., Jackson, D., Silhavy, T.J., Weisemann, J. and Zweig, M. (1983) Open reading frame expression vectors: a general method for antigen production in Escherichia coli using protein fusions to beta-galactosidase. Proceedings of the National Academy of Sciences. 80, 4432–4436.
Xie, J.L., Bohovych, I., Wong, E.O.Y., Lambert, J.-P., Gingras, A.-C., Khalimonchuk, O., Cowen, L.E. and Leach, M.D. (2017) Ydj1 governs fungal morphogenesis and stress response, and facilitates mitochondrial protein import via Mas1 and Mas2. Microbial Cell. 4, 342–361.
Dong J, Zhang Y, Chen Y, Niu X, Zhang Y, Yang C, Wang Q, Li X, Deng X. 2015. Baicalin inhibits the lethality of Shiga-like toxin 2 in mice. Antimicrob Agents Chemother. 2015 Nov;59(11):7054-60. Zhang Y, Qi Z, Liu Y, He W, Yang C, Wang Q, Dong J, Deng X. 2017. Baicalin Protects Mice from Lethal Infection by Enterohemorrhagic Escherichia coli. Front Microbiol. 8:395. Dinda B, Dinda S, DasSharma S, Banik R, Chakraborty A, Dinda M. Therapeutic potentials of baicalin and its aglycone, baicalein against inflammatory disorders. Journal of Medicinal Chemistry 2017, 68–80. O'Brien AD, Newland JW, Miller SF, Holmes RK, Smith HW, Formal SB. 1984. Shiga-Like Toxin-Converting Phages from Eschenrchia coli Strains That Cause Hemorrhagic Colitis or Infantile Diarrhea. Science 226: 694-696. Karve SS, Weiss AA. 2014. Glycolipid binding preferences of Shiga toxin variants. PLoS One. 9(7): e101173.
We appreciate your important suggestion to make our manuscript much more valuable.
If the revised manuscript is still insufficient, could you please give us further comments.
Reviewer 3 Report
This manuscript describes an interesting study of the effect of the phenolic compound Baicalein on Shiga toxin (Stx) toxicity on Vero cells. The study uses crude protein extracts from Stx1 or Stx2 producing enterohaemorragic Escherichia coli (EHEC) and commercially available baicalein and shows that baicalein protects Vero cells from the effect of Stx1 and Stx2. Furthermore the study shows that although transcription of stx1, but not stx2 increases in the presence of baicalein, the protein concentration of these toxins remain unchanged. The authors hypothesize that baicalein acts by interacting with the B-subunit pentamer of the toxins and show by in silico docking stuies that baicalein can form hydrogen bonds with B-subunits of Stx.
Major concerns:
The language of the manuscript is a major concern. The text must be corrected; preferably by a native English speaker.
The effect of baicalein on Stx toxicity is only tested using the MTT proliferation assay. This assay measures metabolic activity of a cell culture. Since the authors cite literature, showing an effect by baicalein on metabolic enzymes it would be valuable to analyze the effects of baicalein in additional and complementary assays.
The discussion is rather confusing and it is not always possible to understand how the different statements relate to the present study. This may be due to the poor language of the manuscript, but this reviewer suggest that the authors make an effort and only discuss the present results in light of relevant literature.
The authors discuss at length that the observed protective effect of baicalein against Stx is due to a possible direct interaction of bacalein and the StxB-pentamer. The discussion is entirely based on docking studies using the crystal structure of the StxB-pentamer only. However, Stx is only cytotoxic as the AB5-holotoxin, and the StxB-pentamer is completely inactive as a toxin. Since it is known from crystal structures of both Stx1 and Stx2 that the StxA subunit protrudes through the “pore” of the StxB-pentamer, and likely occupies at least part of the suggested interaction site, it would be necessary to model the interaction between baicalein and the Stx-holotoxin in order to draw any relevant conclusions. Furthermore, the authors mention that baicalein may affect membrane fluidity, but this is not discussed as a possible mechanism for the protective effect against Stx. It is known that compounds affecting membrane fluidity reduce the cytotoxic effects of Stx by changing the dynamics of the retrograde transport of the toxin from the cell surface to the endoplasmic reticulum.
Minor concerns:
From figure 1 and 2 it is clear that baicalein has a significant effect on the viability (metabolism) of Vero cells, at least at the lower concentration. This fact and the possible reasons for it are not discussed in the manuscript.
Figure 3 is missing the error bars.
Line 107, what is the sign between OD660 and 0.6? This should probably be changed to “=”.
Figure 5 should be improved. Due to the white rings and numbers it is impossible to see the hydrogen bonds and interpret the data. The thin wireframe depiction of the StxB-pentamer without numbering of the amino acids makes it very hard to read anything out of the figure. The figure should be improved to simplify for the readers and allow them to interpret the data.
Lines 203 – 204: The authors state that baicalein did not seem to affect membrane fluidity, but there is no data supporting this statement.
Author Response
Dear reviwer,
Thank you for your kind reviewing our article (Manuscript ID: toxins-569419) entitled ‘Effect of baicalein on the cytotoxicity, production and secretion of Shiga toxins of enterohaemorrhagic Escherichia coli”. According to your comment, we revised the manuscript and made response sheet to answer to each comment from you. If our revisions are still inadequate, your further kind suggestions would be appreciated for improving our revised manuscript.
Comment from you:
This manuscript describes an interesting study of the effect of the phenolic compound Baicalein on Shiga toxin (Stx) toxicity on Vero cells. The study uses crude protein extracts from Stx1 or Stx2 producing enterohaemorragic Escherichia coli (EHEC) and commercially available baicalein and shows that baicalein protects Vero cells from the effect of Stx1 and Stx2. Furthermore the study shows that although transcription of stx1, but not stx2 increases in the presence of baicalein, the protein concentration of these toxins remain unchanged. The authors hypothesize that baicalein acts by interacting with the B-subunit pentamer of the toxins and show by in silico docking stuies that baicalein can form hydrogen bonds with B-subunits of Stx.
Major concerns:
The language of the manuscript is a major concern. The text must be corrected; preferably by a native English speaker.
Answer:
Thank you for your careful reviewing. We check the manuscript again and also asked native speaking colleagues to check it. If our revised manuscript is not enough to be accepted, could you please let us know?
The effect of baicalein on Stx toxicity is only tested using the MTT proliferation assay. This assay measures metabolic activity of a cell culture. Since the authors cite literature, showing an effect by baicalein on metabolic enzymes it would be valuable to analyze the effects of baicalein in additional and complementary assays.
The discussion is rather confusing and it is not always possible to understand how the different statements relate to the present study. This may be due to the poor language of the manuscript, but this reviewer suggest that the authors make an effort and only discuss the present results in light of relevant literature.
The authors discuss at length that the observed protective effect of baicalein against Stx is due to a possible direct interaction of bacalein and the StxB-pentamer. The discussion is entirely based on docking studies using the crystal structure of the StxB-pentamer only. However, Stx is only cytotoxic as the AB5-holotoxin, and the StxB-pentamer is completely inactive as a toxin. Since it is known from crystal structures of both Stx1 and Stx2 that the StxA subunit protrudes through the “pore” of the StxB-pentamer, and likely occupies at least part of the suggested interaction site, it would be necessary to model the interaction between baicalein and the Stx-holotoxin in order to draw any relevant conclusions. Furthermore, the authors mention that baicalein may affect membrane fluidity, but this is not discussed as a possible mechanism for the protective effect against Stx. It is known that compounds affecting membrane fluidity reduce the cytotoxic effects of Stx by changing the dynamics of the retrograde transport of the toxin from the cell surface to the endoplasmic reticulum.
Answer: We appreciate your important suggestion to make our manuscript much more valuable.
If the revised manuscript is still insufficient, could you please give us further comments.
Minor concerns:
From figure 1 and 2 it is clear that baicalein has a significant effect on the viability (metabolism) of Vero cells, at least at the lower concentration. This fact and the possible reasons for it are not discussed in the manuscript.
Answer: Thank you for your careful reviewing. In the discussion, we mentioned the protecive effects of some compounds on Vero cell and compared that of baicalein (P. 7; lines 180-185). We also added for the effects of baicalein on the viability of Vero cell “At 0.13 mmol/L, baicalein significantly reduced the cytotoxicity of Stx1 at concentrations ranging from 0.5 to 33.3 ng/mL, and also that of Stx2 from 2.1 to 533.3 ng/mL”. (P.3; Lines 79-80).
Figure 3 is missing the error bars.
Answer: Thank you for your comment. Because we performed two separate experiments in this. Hence, Figure 3 did not show the error bars.
Line 107, what is the sign between OD660 and 0.6? This should probably be changed to “=”.
Answer: Thank you for your comment. We changed “≒” to “=” according your suggestion.
Figure 5 should be improved. Due to the white rings and numbers it is impossible to see the hydrogen bonds and interpret the data. The thin wireframe depiction of the StxB-pentamer without numbering of the amino acids makes it very hard to read anything out of the figure. The figure should be improved to simplify for the readers and allow them to interpret the data.
Answer: Considering your comment sincerely. We are sorry for this mistake. We have improved in figure 5 and it was attched as below. If our modification is not enough, please let us know.
Figure 5 Docking models of baicalein bound to the pockets of Stx1B and Stx2B pentamers. Hydrogen bonds formed between baicalein and amino acids facing inside the pockets of Stx1B and Stx2B pentamers are enclosed with white circles and numbered. (A) Model showing 2 hydrogen bonds formed between baicalein and side chains of amino acid facing inside the pocket of the Stx1B pentamer at Ser42 and Ser42 of Monomer B. (B) Model showing 1 hydrogen bond formed between baicalein and side chain of amino acid of the Stx2B pentamer at Ser41 of Monomer J.
Lines 203 – 204: The authors state that baicalein did not seem to affect membrane fluidity, but there is no data supporting this statement.
Answer: Thank you for your comment. As you mentioned, although there is no data supporting this statement, we based on the relevant publications to explain for the result of baicalein had no effect on the secretion of Stxs.
Considering your comment sincerely, if our discussion for this statement is not enough, please let us know.
We appreciate your important suggestion to make our manuscript much more valuable.
If the revised manuscript is still insufficient, could you please give us further comments.
Round 2
Reviewer 1 Report
GENERAL COMMENTS
“Baicalein inhibits Stx1 and 2 of EHEC – Effects of baicalein on the cytotoxicity, production and secretion of Shiga toxins of enterohaemorrhagic Escherichia coli” (toxins-569419-peer review-v2) is a revised manuscript submitted following initial blind review by three independent reviewers. The authors are to be commended for their careful attention to each of the reviewers’ comments and suggestions. The manuscript has been incrementally improved by the review process. Overall, however, English language usage is still a concern in this manuscript, and editorial assistance should be offered to the authors prior to publication of the manuscript. The reviewer will not cite all instances of incorrect English language usage, but will offer suggested revisions for some passages in the fourteen specific comments below.
SPECIFIC COMMENTS
Lines 28-35: These revised sentences, while more accurate, are difficult for the reader to follow. Please consider the following suggested changes: “The Stx receptors are glycolipids of the globo-series, of which globotriaosylceramide (Gb3) is the primary receptor found on the surface of vascular endothelial cells and kidney epithelial cells. However, Stx subtypes have been shown to bind with different affinities to multiple glycolipid receptors. Stx1a preferentially binds to Gb3 with detectable binding to globotetraosylceramide (Gb4), whereas Stx2a strongly binds to Gb3 and marginally binds to Gb4 [4]. Stx2e binds to both Gb3 and Gb4, and also to the Forssman glycosphingolipid [5]. The Stx pentameric B subunit interacts with Gb3 and induces membrane invagination, leading to the internalization of the toxin. The internalized Stxs inhibit protein synthesis, leading to cell death. Recently, however, Lee et al. (2016) have reviewed the multifunctionality of Stxs that not only inhibit protein synthesis but also induce apoptosis in different cell types [3].”
Lines 38-40: Change to: “…and non-steroidal anti-inflammatory agents [9].”
Lines 41-42: The clarity of this concluding sentence may be improved by changing to: “Clearly, the demand for the development of novel therapies to prevent or treat EHEC infections is increasing.”
Lines 47-48: The term “inhibited effects” may be interpreted different ways. To improve clarity, consider changing to: “Phytochemicals, including plant polyphenols, have shown inhibitory effects on verocytotoxin-producing E.coli O157:H7 [14].”
Lines 59-60 and elsewhere: As mentioned by another reviewer, it would be best to designate the Stx subtypes used in this study. With this in mind, change the sentence to read “Among the compounds selected by the in silico screening, baicalein reduced the cytotoxicity of Stx1a and Stx2a.” These changes in Stx subtype nomenclature should be made throughout the manuscript.
Lines 60-62: To clarify this concluding sentence, consider changing to: “The effects of baicalein, which showed the strongest inhibitory activity against Stx, were investigated on the protection of Vero cells against Stx, and the production of Stxs by EHEC.”
Lines 65-69: In this reviewer’s opinion, the data shown in Figure S-1 are sufficiently significant to warrant publication in the paper. The data clearly show that among the compounds tested, baicalein is superior in protecting cells from Stx1a cytotoxicity, while protection against Stx2a intoxication is less extensive.
Lines 70-71: Consider changing to “Table 1. Candidate compounds for Stx inhibitors selected from a collection of purified natural products isolated from plants (MEGxp; AnalytiCon Discovery) by docking simulation.”
Lines 73-81 and Figure 1A and 1B: The authors have noted in their response to a previous review of this manuscript that:
“…there are consistent differences in Vero cell viability at the concentration Stxs 0 ng/mL because the cytotoxicity of baicalein effects on Vero cell. According to Zandi et al. 2012 reported that the viability of Vero cell reduce by 50% by the concentration of baicalein at 109 μg/mL. However, baicalein in particular and flavonoids in general were described as less toxic in comparison to other plant compounds (alkaloids). In this study, we would like focus on the effects of baicalein on the cytotoxicity of Stx1 and Stx2 instead of inhibition of baicalein on Vero cell growth.”
In this reviewer’s opinion, this is important information that should be shared with the readers at this point. The reviewer understands that the authors wish to emphasize the capacity of baicalein to reduce Stx-mediated Vero cell cytotoxicity in this experiment, but in the long term, any direct cytotoxic effects mediated by baicalein treatment alone must be considered in the development of its use as a novel therapeutic agent to treat EHEC infections. It is this reviewer’s recommendation that you include a comment similar to the above written response to the first review and include the reference of Zandi et al. (2012) in the manuscript. It is odd that the baicalein cytotoxic effect is greater at the lower dose (0.027 mmol/L) versus the higher dose (0.13 mmol/L) of baicalein. Any explanation for this?
Line 104: To improve clarity, consider changing the subheading to: “Effects of baicalein on production of Stx by EHEC.”
Lines 105-113, lines 115-118 and Figure 3: In response to an earlier review of the manuscript, the authors wrote:
“As we mentioned the use of antibiotics to treat EHEC infections is controversial because the increas of Stx production and secretion lead to the risk of HUS development. So, we performed the experiments to determine the effects of baicalein on transcirption of stx1 and stx2 in compared with negative control (water) and positive control (Mitomycin C – an antibiotic was used to treat bacteria infection). As you mentioned, Is there evidence in the literature that baicalein regulates gene expression in bacteria? According to Cai et al., 2016 have reported that baicalein significantly inhibited the expression of CTX-M-1 mRNA expression in Klebsiella pneumoniae strains.”
Again, this is useful information that the authors should consider sharing with the readers – it tells them why the investigators did the experiment. Consider introducing subsection 2.3 with something like this: “The use of antibiotics to treat EHEC infections is controversial because they can lead to an increase in toxin production and secretion, thereby increasing the risk of HUS development. In addition, baicalein significantly inhibited the expression of CTX-M-1 mRNA expression in Klebsiella pneumoniae strains (Cai et al., 2016). We therefore performed experiments designed to determine the effects of baicalein on stx1a and stx2a transcription compared to negative (water) and positive (mitomycin C) controls.”
Lines 115-118: In an earlier review of the manuscript, the authors were asked the question: Are E. coli O157:H7 strains 33 and 36 isogenic except for toxin production? The authors responded:
“Some E.coli O157:H7 strains produce only one toxin type, etheir Stx1 or Stx2 while others express both Stx1 and Stx2. For each strain: E.coli O157:H7 No.33 (stx1+, stx2-) and E.coli O157:H7 No.36 (stx1-, stx2+) (+) denotes E.coli produce toxin type, (-) denotes E.coli strain do not produce toxin. Sub-type of Stx1 strain 33 was Stx1a, sub-type of Stx2 strain 36 and 184 were Stx2a. As we mentioned as above, these strains isolated by our laboratory and the results have been published.”
The reviewer understands that the strains differ in toxin subtype production, but the question is – do the strains differ at other genetic loci or are they isogenic (identical at all other genetic loci). When I looked up the paper by Miyamoto et al., (Food Control 42: 263-269 [2014]) it doesn’t tell me much about the derivation of the strains. I assume they were isolated from different patients and are, therefore, non-isogenic. As the authors mention in the Discussion section of the manuscript (lines 212-216), there are many genes, both phage-encoded and chromosomally encoded, that may contribute to the expression and secretion of Stxs. The use of strains that are isogenic except for Stx subtype expression would be useful in determining the mechanisms by which baicalein mediates toxin transcriptional changes and protein production. The authors should simply denote if the strains are or are not isogenic.
Lines 157-239: The Discussion section should be carefully reviewed by an editoral assistant before publication.
Line 162: Change “surviving” to survivin.
Author Response
Dear,
Thank you for your kind reviewing our article (Manuscript ID: toxins-569419) entitled ‘Effect of baicalein on the cytotoxicity, production and secretion of Shiga toxins of enterohaemorrhagic Escherichia coli”. According to your comment, we revised the manuscript and made response sheet to answer to each comment from you. If our revisions are still inadequate, your further kind suggestions would be appreciated for improving our revised manuscript.
Comment from you:
GENERAL COMMENTS
“Baicalein inhibits Stx1 and 2 of EHEC – Effects of baicalein on the cytotoxicity, production and secretion of Shiga toxins of enterohaemorrhagic Escherichia coli” (toxins-569419-peer review-v2) is a revised manuscript submitted following initial blind review by three independent reviewers. The authors are to be commended for their careful attention to each of the reviewers’ comments and suggestions. The manuscript has been incrementally improved by the review process. Overall, however, English language usage is still a concern in this manuscript, and editorial assistance should be offered to the authors prior to publication of the manuscript. The reviewer will not cite all instances of incorrect English language usage, but will offer suggested revisions for some passages in the fourteen specific comments below.
Answer: Thank you for your careful reviewing. English language usage in our manscripts have revised by MDPI English editing.
SPECIFIC COMMENTS
Lines 28-35: These revised sentences, while more accurate, are difficult for the reader to follow. Please consider the following suggested changes: “The Stx receptors are glycolipids of the globo-series, of which globotriaosylceramide (Gb3) is the primary receptor found on the surface of vascular endothelial cells and kidney epithelial cells. However, Stx subtypes have been shown to bind with different affinities to multiple glycolipid receptors. Stx1a preferentially binds to Gb3 with detectable binding to globotetraosylceramide (Gb4), whereas Stx2a strongly binds to Gb3 and marginally binds to Gb4 [4]. Stx2e binds to both Gb3 and Gb4, and also to the Forssman glycosphingolipid [5]. The Stx pentameric B subunit interacts with Gb3 and induces membrane invagination, leading to the internalization of the toxin. The internalized Stxs inhibit protein synthesis, leading to cell death. Recently, however, Lee et al. (2016) have reviewed the multifunctionality of Stxs that not only inhibit protein synthesis but also induce apoptosis in different cell types [3].”
Answer: Thank you for your suggestion and advice. We revised lines 28-35 according to your suggestion.
The Stx receptors are glycolipids of the globo-series, of which globotriaosylceramides (Gb3s) is the primary receptor found on the surface of vascular endothelial cells and kidney epithelial cells. However, Stx subtypes have been shown to bind with different affinities to multiple glycolipid receptors. Stx1a preferentially binds to Gb3, with detectable binding to globotetraosylceramide (Gb4) whereas Stx2a strongly binds to Gb3 and marginally binds to Gb4 [3]. Stx2e binds to both Gb3 and Gb4, and also to the Forssman glycosphingolipid [4]. The Stx pentameric B subunit interacts with Gb3 and induces membrane invagination, leading to the internalization of the toxin. The internalized Stxs inhibit protein synthesis, leading to cell death. Recently, however, Lee et al. (2016) reviewed the multifunctionality of Stxs, which not only inhibits protein synthesis but also induces apoptosis in different cell types [5]. (Lines 26-35)
Lines 38-40: Change to: “…and non-steroidal anti-inflammatory agents [9].”
Thank you for your advice. We corrected “…and non-steroidal anti-inflammatory[9].” to “…and non-steroidal anti-inflammatory agents [9].” according to your suggestion. (Line 40)
Lines 41-42: The clarity of this concluding sentence may be improved by changing to: “Clearly, the demand for the development of novel therapies to prevent or treat EHEC infections is increasing.”
Answer: Thank you for your kindness. We rewrote the sentence according to your suggestion “Clearly, the demand for the development of novel therapies to prevent or treat EHEC infections is increasing.” (Lines 41-42)
Lines 47-48: The term “inhibited effects” may be interpreted different ways. To improve clarity, consider changing to: “Phytochemicals, including plant polyphenols, have shown inhibitory effects on verocytotoxin-producing E.coli O157:H7 [14].”
Answer: Thank you for your kindness. We rewrote the sentence according to your suggestion “Phytochemicals, including plant polyphenols, have shown inhibitory effects on verocytotoxin-producing E.coli O157:H7 [14].” (Lines 47-48)
Lines 59-60 and elsewhere: As mentioned by another reviewer, it would be best to designate the Stx subtypes used in this study. With this in mind, change the sentence to read “Among the compounds selected by the in silico screening, baicalein reduced the cytotoxicity of Stx1a and Stx2a.” These changes in Stx subtype nomenclature should be made throughout the manuscript.
Answer: Thank you for your comment. We am sorry designate the Stx subtypes used in this study. We corrected. E.coli O157:H7 No.33 (stx1+, stx2-), E.coli O157:H7 No.36 (stx1-, stx2+), E.coli O157:H7 No.184 (stx1-, stx2+).
Lines 60-62: To clarify this concluding sentence, consider changing to: “The effects of baicalein, which showed the strongest inhibitory activity against Stx, were investigated on the protection of Vero cells against Stx, and the production of Stxs by EHEC.”
Answer: Thank you for your kindness. We rewrote the sentence according to your suggestion “The effects of baicalein, which showed the strongest inhibitory activity against Stx, were investigated on the protection of Vero cells against Stx, and the production of Stxs by EHEC.” (Lines 60-62)
Lines 65-69: In this reviewer’s opinion, the data shown in Figure S-1 are sufficiently significant to warrant publication in the paper. The data clearly show that among the compounds tested, baicalein is superior in protecting cells from Stx1a cytotoxicity, while protection against Stx2a intoxication is less extensive.
Answer: Thank you for your advice.
Lines 70-71: Consider changing to “Table 1. Candidate compounds for Stx inhibitors selected from a collection of purified natural products isolated from plants (MEGxp; AnalytiCon Discovery) by docking simulation.”
Answer: Thank you for your kindness. We rewrote the sentence according to your suggestion. “Table 1. Candidate compounds for Stx inhibitors selected from a collection of purified natural products isolated from plants (MEGxp; AnalytiCon Discovery) by docking simulation.” (Lines 70-72)
Lines 73-81 and Figure 1A and 1B: The authors have noted in their response to a previous review of this manuscript that:
“…there are consistent differences in Vero cell viability at the concentration Stxs 0 ng/mL because the cytotoxicity of baicalein effects on Vero cell. According to Zandi et al. 2012 reported that the viability of Vero cell reduce by 50% by the concentration of baicalein at 109 μg/mL. However, baicalein in particular and flavonoids in general were described as less toxic in comparison to other plant compounds (alkaloids). In this study, we would like focus on the effects of baicalein on the cytotoxicity of Stx1 and Stx2 instead of inhibition of baicalein on Vero cell growth.”
In this reviewer’s opinion, this is important information that should be shared with the readers at this point. The reviewer understands that the authors wish to emphasize the capacity of baicalein to reduce Stx-mediated Vero cell cytotoxicity in this experiment, but in the long term, any direct cytotoxic effects mediated by baicalein treatment alone must be considered in the development of its use as a novel therapeutic agent to treat EHEC infections. It is this reviewer’s recommendation that you include a comment similar to the above written response to the first review and include the reference of Zandi et al. (2012) in the manuscript. It is odd that the baicalein cytotoxic effect is greater at the lower dose (0.027 mmol/L) versus the higher dose (0.13 mmol/L) of baicalein. Any explanation for this?
Answer: Thank you for your suggestion. To be honest, we do not know why the baicalein cytotoxic effect is greater at the lower dose (0.027 mmol/L) versus the higher dose (0.13 mmol/L) of baicalein. Further study is needed to understand this phenomenon.
Line 104: To improve clarity, consider changing the subheading to: “Effects of baicalein on production of Stx by EHEC.”
Answer: Thank you for your suggestion. We rewrote “Effects of baicalein on productivity of Stx by EHEC.” to “Effects of baicalein on production of Stx by EHEC.” (Line 105)
Lines 105-113, lines 115-118 and Figure 3: In response to an earlier review of the manuscript, the authors wrote:
“As we mentioned the use of antibiotics to treat EHEC infections is controversial because the increas of Stx production and secretion lead to the risk of HUS development. So, we performed the experiments to determine the effects of baicalein on transcirption of stx1 and stx2 in compared with negative control (water) and positive control (Mitomycin C – an antibiotic was used to treat bacteria infection). As you mentioned, Is there evidence in the literature that baicalein regulates gene expression in bacteria? According to Cai et al., 2016 have reported that baicalein significantly inhibited the expression of CTX-M-1 mRNA expression in Klebsiella pneumoniae strains.”
Again, this is useful information that the authors should consider sharing with the readers – it tells them why the investigators did the experiment. Consider introducing subsection 2.3 with something like this: “The use of antibiotics to treat EHEC infections is controversial because they can lead to an increase in toxin production and secretion, thereby increasing the risk of HUS development. In addition, baicalein significantly inhibited the expression of CTX-M-1 mRNA expression in Klebsiella pneumoniae strains (Cai et al., 2016). We therefore performed experiments designed to determine the effects of baicalein on stx1a and stx2a transcription compared to negative (water) and positive (mitomycin C) controls.”
Answer: Thank you for your suggestion. We added this information to the revised manuscript.
“In addition, baicalein significantly inhibited the expression of CTX-M-1 mRNA expression in Klebsiella pneumoniae strains [39]. We therefore performed experiments designed to determine the effects of baicalein on stx1 and stx2 transcription compared to negative (water) and positive (MMC) controls.” (Lines 211-214)
Lines 115-118: In an earlier review of the manuscript, the authors were asked the question: Are E. coli O157:H7 strains 33 and 36 isogenic except for toxin production? The authors responded:
“Some E.coli O157:H7 strains produce only one toxin type, etheir Stx1 or Stx2 while others express both Stx1 and Stx2. For each strain: E.coli O157:H7 No.33 (stx1+, stx2-) and E.coli O157:H7 No.36 (stx1-, stx2+) (+) denotes E.coli produce toxin type, (-) denotes E.coli strain do not produce toxin. Sub-type of Stx1 strain 33 was Stx1a, sub-type of Stx2 strain 36 and 184 were Stx2a. As we mentioned as above, these strains isolated by our laboratory and the results have been published.”
The reviewer understands that the strains differ in toxin subtype production, but the question is – do the strains differ at other genetic loci or are they isogenic (identical at all other genetic loci). When I looked up the paper by Miyamoto et al., (Food Control 42: 263-269 [2014]) it doesn’t tell me much about the derivation of the strains. I assume they were isolated from different patients and are, therefore, non-isogenic. As the authors mention in the Discussion section of the manuscript (lines 212-216), there are many genes, both phage-encoded and chromosomally encoded, that may contribute to the expression and secretion of Stxs. The use of strains that are isogenic except for Stx subtype expression would be useful in determining the mechanisms by which baicalein mediates toxin transcriptional changes and protein production. The authors should simply denote if the strains are or are not isogenic.
Answer: Thank you for your careful reviewing. We are really sorry for confusion in the previous response. We corrected E.coli O157:H7 No.33 (stx1+, stx2-), E.coli O157:H7 No.36 (stx1-, stx2+), E.coli O157:H7 No.184 (stx1-, stx2+). Sub-type of these strains in detail will be published in another study of our laboratory.
As you mentioned, the use of strains that are isogenic except for Stx subtype expression would be useful in determining the mechanisms by which baicalein mediates toxin transcriptional changes and protein production. However, E. coli O157:H7 strains No.33 and 36 used in this study are not isogenic beacause they are different E.coli O157:H7 strains.
If the revised manuscript is still insufficient, could you please give us further comments.
Lines 157-239: The Discussion section should be carefully reviewed by an editoral assistant before publication.
Answer: Thank you for your comment. If our revisions are still inadequate, your further kind suggestions would be appreciated for improving our revised manuscript.
Line 162: Change “surviving” to survivin.
Answer: Thank you very much. We corrected “surviving” to “survivin” according to your suggestion. (Line 163)
Again, we appreciate your important suggestion to make our manuscript much more valuable.
If the revised manuscript is still insufficient, could you please give us further comments.
Reviewer 2 Report
Dear Editor and Authors,
The authors have added the required text that this reviewer requested. The text was further edited to improve the grammar and flow. It is now easier to read. They elected not to add introductory text stating that the temperate lambdoid phages infecting the E. coli host are responsible for the production of Shiga toxins. This reviewer feels this weakens the manuscript, but it is the authors’ choice.
The newly added text has a disturbing number of typographical errors. The authors need to correct these. Examples from lines 31-34 are listed below. That list in not a complete list. The authors need to carefully read the manuscript to remove any other errors that are present.
The authors need to carefully review the manuscript to ensure that typographical errors, such as the on Line 208, are removed.
Line 31 multifunctionality is misspelled (multifuncationality)
Line 31 also is misspelled (also)
Lines 30-32 reads as follows:
protein synthesis, leading to cell death. Recently, however, Lee et al. (2016) have also presented on the multifuncationality of Stxs that not only inhibits protein synthesis but aslo induced apoptosis in different cell types [3].
The Lee et al. (2006) looks like a reference citation from another journal. It should be adjusted to meet the journal standards.
This construct is also found six other times in lines 164, 188, 204, 216, 225, and 257.
Lines 33-34: Stx1a preferential binding to Gb3 and only detectably binds with Gb4 whereas Stx2a strongly binds to Gb3 and marginally bind to globotetraosylceramide (Gb4) [4].
preferential should be preferentially.
What does “only detectably binds” mean? Is the same as marginally bind or does it mean binds to a lesser extent to Gb4 than to Gb3. It is not clear what the authors mean.
Line 208 inhibition is misspelled (inhibiton)
Citations
The references need to be formatted correctly. The author list should be complete and the initials of authors given names should have periods. He title of the manuscript should be properly capitalized. The journal names should be italicized, properly capitalized, and abbreviated. The year of publication should be bold followed by a comma and a space, then the volume number should be italicized and followed by a
Reference 30 has 30 in the author list.
Author Response
Dear reviewer,
Thank you for your kind reviewing our article (Manuscript ID: toxins-569419) entitled ‘Effect of baicalein on the cytotoxicity, production and secretion of Shiga toxins of enterohaemorrhagic Escherichia coli”. According to your comment, we revised the manuscript and made response sheet to answer to each comment from you. If our revisions are still inadequate, your further kind suggestions would be appreciated for improving our revised manuscript.
Comment from you:
The authors have added the required text that this reviewer requested. The text was further edited to improve the grammar and flow. It is now easier to read. They elected not to add introductory text stating that the temperate lambdoid phages infecting the E. coli host are responsible for the production of Shiga toxins. This reviewer feels this weakens the manuscript, but it is the authors’ choice.
Answer: Our revised manuscript still inadequate. Thank you for your kindness.
The newly added text has a disturbing number of typographical errors. The authors need to correct these. Examples from lines 31-34 are listed below. That list in not a complete list. The authors need to carefully read the manuscript to remove any other errors that are present.
The authors need to carefully review the manuscript to ensure that typographical errors, such as the on Line 208, are removed.
Line 31 multifunctionality is misspelled (multifuncationality)
Line 31 also is misspelled (also)
Lines 30-32 reads as follows:
protein synthesis, leading to cell death. Recently, however, Lee et al. (2016) have also presented on the multifuncationality of Stxs that not only inhibits protein synthesis but aslo induced apoptosis in different cell types [3].
The Lee et al. (2006) looks like a reference citation from another journal. It should be adjusted to meet the journal standards.
Answer: Thank you for your comment. We rewrote in the revised manuscript.
“Recently, however, Lee et al. (2016) have reviewed the multifunctionality of Stxs that not only inhibits protein synthesis but aslo induce apoptosis in different cell types [5]”. Lines 34-35.
This construct is also found six other times in lines 164, 188, 204, 216, 225, and 257.
Lines 33-34: Stx1a preferential binding to Gb3 and only detectably binds with Gb4 whereas Stx2a strongly binds to Gb3 and marginally bind to globotetraosylceramide (Gb4) [4].
preferential should be preferentially.
What does “only detectably binds” mean? Is the same as marginally bind or does it mean binds to a lesser extent to Gb4 than to Gb3. It is not clear what the authors mean.
Answer: Thank you for your careful reviewing. We rewrote in the revised manuscript. If the revised manuscript is still insufficient, could you please give us further comments.
“The Stx receptors are glycolipids of the globo-series, of which globotriaosylceramides (Gb3s) is the primary receptor found on the surface of vascular endothelial cells and kidney epithelial cells. However, Stx subtypes have been shown to bind with different affinities to multiple glycolipid receptors. Stx1a preferentially binds to Gb3, with detectable binding to globotetraosylceramide (Gb4) whereas Stx2a strongly binds to Gb3 and marginally binds to Gb4 [3]. Stx2e binds to both Gb3 and Gb4, and also to the Forssman glycosphingolipid [4]. The Stx pentameric B subunit interacts with Gb3 and induces membrane invagination, leading to the internalization of the toxin. The internalized Stxs inhibit protein synthesis, leading to cell death. Recently, however, Lee et al. (2016) reviewed the multifunctionality of Stxs, which not only inhibits protein synthesis but also induces apoptosis in different cell types [5].” Lines 26-35
Line 208 inhibition is misspelled (inhibiton)
Answer: Thank you for your careful reviewing. We corrected “inhibiton” to “inhibition”. Line 208
Citations
The references need to be formatted correctly. The author list should be complete and the initials of authors given names should have periods. He title of the manuscript should be properly capitalized. The journal names should be italicized, properly capitalized, and abbreviated. The year of publication should be bold followed by a comma and a space, then the volume number should be italicized and followed by a
Reference 30 has 30 in the author list.
Answer: Thank you for your comments. We resived the references in the revised manuscript.
Zhang Y, Qi Z, Liu Y, He W, Yang C, Wang Q, et al. Baicalin protects mice from lethal infection by enterohemorrhagic Escherichia coli. Frontiers in Microbiology. 2017, 8, 2–7. [CrossRef]. Lines 445-447
Again, we appreciate your important suggestion to make our manuscript much more valuable.
If the revised manuscript is still insufficient, could you please give us further comments.
Reviewer 3 Report
Dear Authors,
Thank you for the improvements you have done to the manuscript. There are however still a few concerns:
The editing of the text has made it easier to read the manuscript, but it still needs further improvement. I am not going to comment on all problems, but as far as I can see, the first linguistic error occur in the second sentence of the abstract (lines 7 – 9) and then it continues on lines 13, 15, 16, 23, 25, and throughout the manuscript.
I have a major problem with the presentation of Figure 3. I do not agree with the authors that the error bars showing the variation between the replicates should not be shown due to few independent experiments. There are two solutions to this “problem”: 1) perform more experiments, or 2) show all data points so that the reader can make his/her own interpretation of the data.
There is also still a concern regarding figure 5: The circling of the hydrogen bonds is really not needed. It occludes the only important part of the figure and actually hides the side chains that baicalein interacts with. It would be enough to number the bonds with the white numbers. I am also curious, why does baicalein interact with a specific subunit of the pentamer? Can it interact with the corresponding amino acid of all subunits with equally low energy? If not, why not? There is no discussion about this.
The results are relevant for initial analysis of novel drugs to treat EHEC infections. The authors show that baicalein has an effect on toxicity of Shiga toxins. In addition it is shown that baicalein does not seem to induce expression or secretion of Stx, which is important considering the connection between antibiotic use and increased release of toxin with subsequent avderse effects on disease progression. However, as I previously mentioned the results and discussion do not really form a coherent story because the discussion is confusing. Since this was not addressed in the revision I will try to clarify what I mean below:
There is a lot of references to different articles, but it is not always clear how the authors want to relate the findings in these papers with the findings in the current manuscript. For example:
Lines 157 – 164: The authors mention that baicalein have biological effects on Prokaryotic and Eukaryotic organisms as well as viruses. It is unclear how this is related to the effect on Stx toxicity. The authors do not discuss this, just mention that it has been reported.
In the discussion the authros mention that the function of baicalein is: inducer of apoptosis, inhibitor of different enzymes, is antioxidant, neuroprotective, antibacterial, antiviral, antifungal, inhibits biofilms, protects against UV radiation, reduce proliferation of Cancer cells (but it is safe for human consumption, even though Figures 1 and 2 clearly show a negative effect on cell viability).
Despite the many functions of baicalein, the authors try to explain the effect on Stx toxicity with their in silico modeling data on the interaction between baicalein and the StxB pentamer. Although this may be a potential explanation it is pure speculation:
1) the modeling was done on the crystal structure of the StxB-pentamers only, not the holotoxin. 2) since it has been shown that baicalein can inhibit different enzymes, why do the authors not consider that it is actually the enzymatic activity of Stx that is inhibited? 3) the authors claim that the baicalein-Stx interaction interfers with the interaction of Stx with the receptor on the cell surface, but there is no data supporting this claim. Does baicalein affect the toxin association with the cells? 4) since baicalein binds in the pocket of the STxB-pentamer, how does this affect the stability of the toxin complex? Is the A subunit still associated with the B-pentamer? 5) The authors speculate that membrane fluidity is changed by baicalein and can have an effect. This is pure speculation, and that is fine, but the authors do not discuss how this may affect the retrograde transport of the toxin. Is this peraps the real effect of baicalein? The authros even cite a paper showing an effect of baicalein on lysosomes. These things are not discussed in any way, only mentioned.
Supplementary data: The figure is informative and adds value to the results, however the figure (at least in my document) is of low quality. The x-axis title is too close to the x-axis.
Author Response
Dear reviewer,
Thank you for your kind reviewing our article (Manuscript ID: toxins-569419) entitled ‘Effect of baicalein on the cytotoxicity, production and secretion of Shiga toxins of enterohaemorrhagic Escherichia coli”. According to your comment, we revised the manuscript and made response sheet to answer to each comment from you. If our revisions are still inadequate, your further kind suggestions would be appreciated for improving our revised manuscript.
Comment from you:
Thank you for the improvements you have done to the manuscript. There are however still a few concerns:
The editing of the text has made it easier to read the manuscript, but it still needs further improvement. I am not going to comment on all problems, but as far as I can see, the first linguistic error occur in the second sentence of the abstract (lines 7 – 9) and then it continues on lines 13, 15, 16, 23, 25, and throughout the manuscript.
Answer: Thank you for your careful reviewing. English language usage in our manscripts have revised by MDPI English editing.
I have a major problem with the presentation of Figure 3. I do not agree with the authors that the error bars showing the variation between the replicates should not be shown due to few independent experiments. There are two solutions to this “problem”: 1) perform more experiments, or 2) show all data points so that the reader can make his/her own interpretation of the data.
Answer: Thank you for kindnes suggestion. As you mentioned, show all data points so that the reader can make his/her own interpretation of the data. We revised Figure 3 according to your suggestion.
There is also still a concern regarding figure 5: The circling of the hydrogen bonds is really not needed. It occludes the only important part of the figure and actually hides the side chains that baicalein interacts with. It would be enough to number the bonds with the white numbers. I am also curious, why does baicalein interact with a specific subunit of the pentamer? Can it interact with the corresponding amino acid of all subunits with equally low energy? If not, why not? There is no discussion about this.
Answer: Thank you for your kind suggestion. As you mentioned, we revised Figure 5 again.
Figure 5 Docking models of baicalein bound to the pockets of Stx1B and Stx2B pentamers. Hydrogen bonds formed between baicalein and amino acids facing inside the pockets of Stx1B and Stx2B pentamers are white numbers. (A) Model showing two hydrogen bonds formed between baicalein and side chains of amino acid facing inside the pocket of the Stx1B pentamer at Ser42 and Ser42 of Monomer B. (B) Model showing one hydrogen bond formed between baicalein and side chain of amino acid of the Stx2B pentamer at Ser41 of Monomer J.
In this study, the docking model showed that baicalein interact with a specific subunit of the pentamer. The mechanism of this interaction is still not clear. However, baicalein can interact with the corresponding amino acid of other subunits with equally low energy. Further study, we will try to clarify this issue.
The results are relevant for initial analysis of novel drugs to treat EHEC infections. The authors show that baicalein has an effect on toxicity of Shiga toxins. In addition it is shown that baicalein does not seem to induce expression or secretion of Stx, which is important considering the connection between antibiotic use and increased release of toxin with subsequent avderse effects on disease progression. However, as I previously mentioned the results and discussion do not really form a coherent story because the discussion is confusing. Since this was not addressed in the revision I will try to clarify what I mean below:
There is a lot of references to different articles, but it is not always clear how the authors want to relate the findings in these papers with the findings in the current manuscript. For example:
Lines 157 – 164: The authors mention that baicalein have biological effects on Prokaryotic and Eukaryotic organisms as well as viruses. It is unclear how this is related to the effect on Stx toxicity. The authors do not discuss this, just mention that it has been reported.
Answer: Thank you for your careful reviewing. We rewrote lines 157-164 to show clarity of the mention that baicalein have biological effects on Prokaryotic and Eukaryotic organisms as well as viruses related to the effect on Stx.
“Furthermore, baicalin, a parental compound of bacalein, significantly reduced the activity of Stx2 which induced lethality in mice [29]. The protection effects of baicalin on the cell against EHEC infection were clarified by Zhang et al. (2017) [30]. To the best of our knowledge, however, there are currently no reports on the inhibition effect of baicalein on the cytotoxicity of both Stx1 and Stx2..” (Lines 165-169)
In the discussion the authros mention that the function of baicalein is: inducer of apoptosis, inhibitor of different enzymes, is antioxidant, neuroprotective, antibacterial, antiviral, antifungal, inhibits biofilms, protects against UV radiation, reduce proliferation of Cancer cells (but it is safe for human consumption, even though Figures 1 and 2 clearly show a negative effect on cell viability).
Despite the many functions of baicalein, the authors try to explain the effect on Stx toxicity with their in silico modeling data on the interaction between baicalein and the StxB pentamer. Although this may be a potential explanation it is pure speculation:
1) the modeling was done on the crystal structure of the StxB-pentamers only, not the holotoxin. 2) since it has been shown that baicalein can inhibit different enzymes, why do the authors not consider that it is actually the enzymatic activity of Stx that is inhibited? 3) the authors claim that the baicalein-Stx interaction interfers with the interaction of Stx with the receptor on the cell surface, but there is no data supporting this claim. Does baicalein affect the toxin association with the cells? 4) since baicalein binds in the pocket of the STxB-pentamer, how does this affect the stability of the toxin complex? Is the A subunit still associated with the B-pentamer? 5) The authors speculate that membrane fluidity is changed by baicalein and can have an effect. This is pure speculation, and that is fine, but the authors do not discuss how this may affect the retrograde transport of the toxin. Is this peraps the real effect of baicalein? The authros even cite a paper showing an effect of baicalein on lysosomes. These things are not discussed in any way, only mentioned.
Answer: Thank you for your suggestion. As you mentioned, there are many functions of baicalein to consider the effects of baicalein on the cytotoxicity of Stxs. In this study, however, we have only determined the interaction between baicalein and StxB pentamer by docking models. Our further researchs will determine whether baicalein can inhibit different enzymes.
We mentioned that the baicalein-Stx interaction interfers with the interaction of Stx with the receptor on the cell surface. As shown in Figure 1 and 2, our think that the findings in this study showed that the baicalein-Stx interaction probably interfers with the interaction of Stx with the receptor on the cell surface. Furthermore, Figure S-1 also showed that among the compounds tested, baicalein is superior in protecting cells from Stx1 cytotoxicity, while protection against Stx2 intoxication is less extensive.
In this study, there is not enough evidence to know exactly whether or not the baicalein-the pocket of the StxB-pentamer interaction affect the stability of the toxin complex. Therefore, further study is needed to clarify this issue.
Thank you for your careful reviewing. To easier to read, we rewrote lines 219-228 to the revised manuscript.
“In our study, treatment with MMC strongly increased the transcription of both stx1 and stx2, while baicalein slightly increased the transcription of stx1 but not stx2 (Figure 3). Stx production of EHEC has been reported to be controlled by several factors, such as growth phase, reactive oxygen species, quorum sensing, H2O2 and neutrophils [40–42]. Specifically, Stx production by EHEC increase at the stationary growth [40,42]. It is also known that Stx production was regulated by phage through the amplification of gene copy number and toxin release [43]. Wagner et al. (2002) suggested that damage of bacterial cell could lead to the release of Stx by the absence of phage-mediated lysis in E.coli O26:H19 [44]. Moreover, baicalein inhibited growth of Staphylococcus aureus by the aggregation of bacterial cells and damaging the bacterial cell membrane [45]. In addition, it has also been reported that the increase of membrane fluidity was associated with the increase of Stx secretion in E.coli O157:H7 [46]. Wu et al. (2013) reported that baicalein exhibited antibacterial activity against E.coli by lowering membrane fluidity of the cells [47]. Together the above findings, the results in this study suggest that baicalein did not affect the growth and membrane fluidity of E.coli O157:H7 at 0.38 mmol/L, though it reduced cytotoxicity of Stx and protected Vero cells at the lower concentration (0.13 mmol/L). However, our data showed that baicalein had no significant effects on the secretion of both Stx1 and Stx2 (Figure 4). This is an advantage of using baicalein to treat EHEC infections over antibiotics.” (Lines 214-230)
Supplementary data: The figure is informative and adds value to the results, however the figure (at least in my document) is of low quality. The x-axis title is too close to the x-axis.
Answer: Thank you for your comment. We revised Figure S-1.
Supplementary Figure S-1. Effects of polyphenols on the cytotoxicity of Stx1 and Stx2. Stx1 (A) and Stx2 (B) preparations containing Stx1 and Stx2 at 12.5 and 50 mg/L, respectively were mixed without (None) or with 100 mg/L of each of the polyphenols and incubated incubated at 37oC for 1 h. After the incubation, the mixture was added to the culture of Vero cells. Cell viability was determined by using MTT Cell Proliferation Assay after cultivation at 37oC for 24. The values are the average of two separate experiments.
Again, we appreciate your important suggestion to make our manuscript much more valuable.
If the revised manuscript is still insufficient, could you please give us further comments.
Round 3
Reviewer 3 Report
Dear Authors,
Thank you for the new version of the manuscript. This is a significantly improved version.
The text is still in need of editing to correct mainly grammatical errors.
In addition, I have a few comments and I suggest the following changes:
Line 13: “However, baicalein had no effect on production or secretion of Stx1 or Stx2.“
Line 15 – 18: “The results demonstrated that inhibitory activity of baicalein against the cytotoxicity of both Stx1 and Stx2 might be due to the formation of a binding structure inside the pocket of the Stx1B and Stx2B pentamers.”
Line 40-41: I do not understand this sentence. It is very confusing. When is there “an increase in EHEC isolated from patients”, and what does “the variety antimicrobial resistance mean”? and how do the two relate to each other?
Line 77-78: “The viability of Vero cells decreased with increasing concentration of Stx1 and Stx2, both in the absence and presence of baicalein”.
Line 78-79: “However, the cytotoxicity of both Stx1 and Stx2 was significantly reduced (P < 0.01) by the pretreatment with baicalein (Figure 1A, B).”
Line 90: “2.2. Protective effect of bacalein on Vero cells against Stx”
Line 210 – 212: I do not understand the sentence. It seems like three different sentences are just jumbled together without giving any real meaning.
Author Response
Dear reviewer,
Thank you for your kind reviewing our article (Manuscript ID: toxins-569419) entitled ‘Effect of baicalein on the cytotoxicity, production and secretion of Shiga toxins of enterohaemorrhagic Escherichia coli”. According to your comment, we revised the manuscript and made response sheet to answer to each comment from you.
The comments from you:
Thank you for the new version of the manuscript. This is a significantly improved version.
The text is still in need of editing to correct mainly grammatical errors.
Answer: Thank you for your careful reviewing. We checked the manuscript again. Thank you very much.
In addition, I have a few comments and I suggest the following changes:
Line 13: “However, baicalein had no effect on production or secretion of Stx1 or Stx2.“
Answer: Thank you for your suggestion. We changed “However, baicalein had no effects on the secretion of both extracellular and intracellular Stxs production” to “However, baicalein had no effect on production or secretion of Stx1 or Stx2” (Line 13) according to your suggestion.
Line 15 – 18: “The results demonstrated that inhibitory activity of baicalein against the cytotoxicity of both Stx1 and Stx2 might be due to the formation of a binding structure inside the pocket of the Stx1B and Stx2B pentamers.”
Answer: Thank you for your advice. We revised “The results demonstrate that inhibitory activity of baicalein against the cytotoxicity of both Stx1 and Stx2 might be the cause of the formation of a binding structure inside the pocket of the Stx1B and Stx2B pentamers” to “The results demonstrated that inhibitory activity of baicalein against the cytotoxicity of both Stx1 and Stx2 might be due to the formation of a binding structure inside the pocket of the Stx1B and Stx2B pentamers” (Lines 15-17).
Line 40-41: I do not understand this sentence. It is very confusing. When is there “an increase in EHEC isolated from patients”, and what does “the variety antimicrobial resistance mean”? and how do the two relate to each other?
Answer: Thank you for your comment. We rewrote this sentence as follows:
“In addition, Hiroi et al. (2012) indicated that EHEC strains isolated from humans have shown increased resistance to one or variety antibiotics [10].” (Lines 41-43)
Line 77-78: “The viability of Vero cells decreased with increasing concentration of Stx1 and Stx2, both in the absence and presence of baicalein”.
Answer: Thank you for your suggestion. We correction “The viability of Vero cells decreased with the increase in the concentration of Stx1 and Stx2 in the absence or presence of baicalein” to “The viability of Vero cells decreased with increasing concentration of Stx1 and Stx2, both in the absence and presence of baicalein” (Lines 79-80) according your suggestion.
Line 78-79: “However, the cytotoxicity of both Stx1 and Stx2 was significantly reduced (P < 0.01) by the pretreatment with baicalein (Figure 1A, B).”
Answer: Thank you for your kind suggestion. We revised “However, the cytotoxicity of both Stx1 and Stx2 was significantly reduced (P < 0.01) by the pretreated with baicalein (Figure 1A,B” to “However, the cytotoxicity of both Stx1 and Stx2 was significantly reduced (P < 0.01) by the pretreatment with baicalein (Figure 1A, B).” (Lines 80-81).
Line 90: “2.2. Protective effect of bacalein on Vero cells against Stx”
Answer: Thank you very much. We changed “Effects protective of baicalein on Vero cells against Stx” to “Protective effect of bacalein on Vero cells against Stx” (Line 92) according to your suggestion.
Line 210 – 212: I do not understand the sentence. It seems like three different sentences are just jumbled together without giving any real meaning.
Answer: Thank you for your suggestion. We rewrote the sentences in Lines 202-212 as follows:
“The inhibition effects of baicalein on the cytotoxicity of Stx were determined (Figure 1 and Figure 2). As the results in Figure 1 and 2 show, there was a decrease in inhibition of bacalein at 0.027 mmol/L in Figure 2 compared to the results in Figure 1. One of possible explanations for this could be that the binding of Stx to the surface of Vero cell was more preferential than the binding of baicalein to StxB pentamers.” (Lines 204-208)